# OR-PRM: A Process Reward Model for Algorithmic Problem in Operations Research

**Yilin Wang**[1,*]**, Heng Zhou**[1,*]**, Dongxing Mao**[2]**, Linjie Li**[3]**, Jingru Tan**[2]**,**
**Haochen Han**[4,†]**, Zhengyuan Yang**[3]**, Alex Jinpeng Wang**[2,†]**, Min Li**[2]

{yilin.wang, hengzhou}@zju.edu.cn, m962479949@gmail.com,
{tanjingru, jinpengwang, liminmin}@csu.edu.cn,
{lindsey.li, zhengyang}@microsoft.com, hhc2077@outlook.com

[1]Zhejiang University, [2]Central South University,
[3]Microsoft, [4]Pengcheng Laboratory

[*]Equal Contribution, [†]Corresponding Authors

## Abstract

Large language models (LLMs) with Process Reward Models (PRMs) have shown strong reasoning ability, yet their potential in Operations Research (OR) remains unexplored. We present the first PRM tailored for OR, but find that directly training on mainstream datasets yields surprisingly weak performance. To understand this gap, we conduct a systematic analysis and identify the primary bottleneck: the datasets themselves, where over 30% of annotations are severely flawed. To overcome these limitations, we first collect all existing synthetic datasets and apply a carefully designed filtering pipeline to construct a high-quality seed dataset. Building upon this seed, we then build OR-ProcessQA, the first large-scale dataset for OR with step-by-step supervision, where diverse solution pathways are generated via Monte Carlo Tree Search (MCTS) and each step is validated for logical consistency by GPT-4o. Building on this foundation, we train OR-PRM, the first Process Reward Model in the OR domain, designed to evaluate and guide reasoning at every step rather than only the final outcome. Together, these advances enable OR-PRM to substantially improve LLMs reasoning capability, achieving a maximum absolute improvement of 12.5% over the base model in Best-of-N settings, and highlighting the power of process-oriented supervision for reliable problem solving in operations research.

## 1 Introduction

Large Language Models (LLMs) DeepSeek-AI (2024); Yang et al. (2025a) have recently demonstrated strong reasoning ability, largely attributed to post-training methods such as reinforcement learning and Process Reward Models (PRMs). Their rapid progress is evident across challenging domainsfor instance, GPT-5 has already surpassed all human competitors in the 2025 ICPC World Finals OpenAI (2025), a notoriously difficult zero-shot programming contest. These advances suggest that LLMs are no longer merely fluent generators, but are evolving into powerful engines for rigorous problem solving.

Operations Research (OR) provides an especially compelling testbed for such reasoning, as it involves modeling and solving complex real-world decision-making problems using mathematical optimization, simulation, and analytical methods to efficiently allocate scarce resources and maximize performance within constrained systems. Solving OR problems demands not only correctness in the final answer, but also step-by-step logical consistencya natural match for PRMs, which are designed to explicitly evaluate the correctness of intermediate steps. At first glance, it seems natural to expect PRMs to excel in OR just as they do in mathematics or programming.

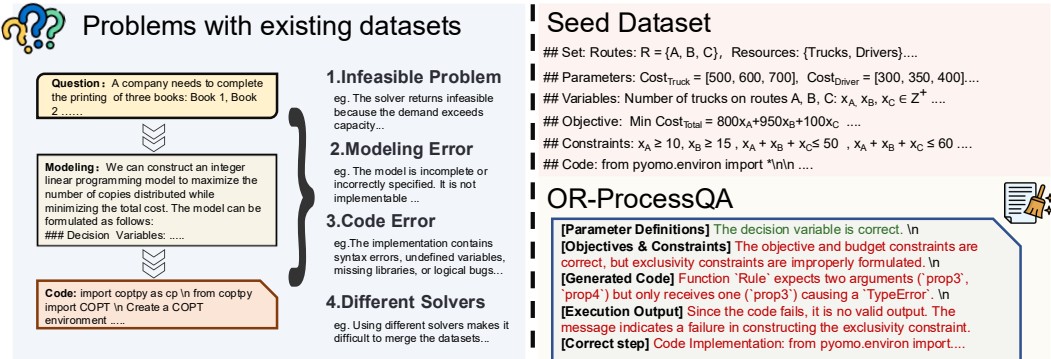

Figure 1: **Noisy Data (left) vs. Our Data (right)**. The left panel illustrates common issues in existing datasets, such as infeasible problems, modeling errors, and coding defects. The right panel showcases our well-structured seed data, which serves as the foundation for our OR-ProcessQA dataset, characterized by step-by-step solutions with explicit correctness labels and ground-truth corrections.

Yet this expectation does not hold. When we developed the first PRM tailored for OR, its performance was far weaker than anticipated, even with state-of-the-art LLM backbones. Our analysis shows that the main obstacle is data quality, since existing OR datasets are alarmingly unreliable. In the Industry OR dataset, even more than 30% of the samples contain serious errors in the final answer, and as with other datasets, many include incomplete or noisy reasoning steps (Figure 1).In one dataset, even, More than 30% of the samples contain serious errors in the final answer, and many include incomplete or noisy reasoning steps (Figure 1). This noise makes it extremely difficult for PRMs to learn faithful reasoning, leading to solutions that look plausible but often violate hidden constraints or break logical consistency.

To overcome these challenges, we first curated a high-quality seed dataset through a rigorous three-stage filtering pipeline. Building on this foundation, we combined MCTS for solution exploration with GPT-4o for fine-grained step-wise annotation, generating hundreds of thousands of problem-solution trajectories. After strict consistency checks, this process yielded OR-ProcessQA, the first large-scale OR dataset with reliable step-level supervision for training PRM.

Leveraging this resource, we developed OR-PRM, the first Process Reward Model tailored for Operations Research. Unlike conventional PRMs that collapse reasoning quality into a single scalar score, OR-PRM delivers structured feedback by categorizing errors and offering targeted corrections. This design enables it to evaluate not only the correctness of final answers but also the validity of every intermediate step. By distinguishing between correct code, incorrect yet runnable code, and non-runnable code, OR-PRM provides actionable guidance for refinement. Our experiments demonstrate that such feedback substantially improves the logical consistency and rule-following behavior of LLMs, marking an important step toward trustworthy decision-making in OR applications.

Overall, our contributions are three-fold: ① We introduce **OR-PRM**, the *first Process Reward Model tailored for Operations Research*, trained to evaluate and guide reasoning at every step rather than relying solely on final answers. ② We curate a high-quality **seed dataset** by filtering existing synthetic OR data, and further expand it with MCTS exploration and GPT-4o annotations into **OR-ProcessQA**, *the first OR dataset with reliable step-level correctness labels for training PRM*. ③ We empirically demonstrate that process-oriented supervision with OR-PRM substantially improves the logical reliability and correctness of LLMs in OR tasks (e.g., achieving average 12.5% accuracy gain on six benchmarks), paving the way toward trustworthy decision-making in real-world applications.

## 2 RELATED WORK

**LLMs for Operations Research**  The remarkable capabilities of LLMs in natural language understanding and complex reasoning have propelled their application in operations research recently. A core challenge lies in effectively translating these naturally described optimization problems into

precise mathematical models that solvers can process. Current academic exploration primarily follows two technical paths Xiao et al. (2025): One path involves reasoning-enhanced methods, which guide general-purpose LLMs in modeling through carefully designed prompts. Examples include X-of-Thought approaches (e.g., the tree-search reasoning employed by Autoformulation Astorga et al. (2025)) and Multi-Expert system (e.g., Chain-of-Experts Xiao et al. (2024) and OptiMUS Ahma-diTeshnizi et al. (2024)). The second path focuses on domain-specific fine-tuning, where models are fine-tuned on specialized datasets to enhance their professional capabilities. Studies such as ORLM Huang et al. (2025a) and LLaMoCo Ma et al. (2024) have demonstrated that fine-tuned models can outperform general-purpose LLMs like GPT-4. Building on this, the LLMOPT Jiang et al. (2025) further advances this direction by introducing the five-element formulation as a universal problem definition paradigm and employing Kahneman-Tversky Optimization (KTO) for model alignment, improving the model's generalization ability.

**Data Synthesis for Operations Research** However, both technical paths above are highly dependent on high-quality datasets. Consequently, researchers have begun exploring data synthesis techniques, broadly categorized into problem-centric and model-centric approaches Xiao et al. (2025). The former, exemplified by OR-Instruct Huang et al. (2025a), augments data by modifying existing problems. The latter prioritizes generating models first and then inversely constructing problem descriptions, thereby offering better control over difficulty and correctness. For instance, the Re-Socratic Yang et al. (2025b) method generates problems inversely from formalized proofs, while OptiMath Lu et al. (2025) and MILP-Evolve Li et al. (2025) generate directly from model code or types. Concurrently, the academic community has released several evaluation benchmarks, including NL4Opt, MAMO, and IndustryOR. Yet, recent studies have uncovered a surprisingly high error rate in these widely used benchmarks (with some datasets exhibiting error rates exceeding 50%) Xiao et al. (2025), severely compromising the reliability of evaluations. Addressing this bottleneck of data quality, this study innovatively clean and construct a batch of high-quality optimization modeling data, laying a solid foundation for training and evaluating more reliable optimization models.

**Process Reward Models** Process Reward Models Cobbe et al. (2021); He et al. (2024); Zhang et al. (2025b;a) provide process-level supervision by scoring intermediate reasoning steps, guiding models to reason step-by-step with improved logical consistency and accuracy. Building on this capability, PRMs have been successfully applied to Best-of-N sampling Wang et al. (2025) and offline data selection Xie et al. (2023), significantly improving reasoning quality and model optimization. Representative works such as Skywork-PRM He et al. (2024) and Qwen2.5-Math-PRM Zhang et al. (2025b) combine human annotations with synthetic rewards to evaluate performance across mathematics, science, and programming domains. They often fail on out-of-distribution reasoning. Zhu et al. (2025) address this with RetrievalPRM, a process reward model using question-and step-level retrieval to improve generalization. Beyond general domains, PRMs are also being extended to vertical domains; for instance, Fin-PRM Zhou et al. (2025) adapts PRMs to finance with trajectory-aware, domain-specialized reward modeling. Applying PRM to vertical domains requires domain-specific knowledge; therefore, we synthesized dataset and conducted training tailored to the characteristics of the Operation Research.

## 3 METHODOLOGY

Our method tackles the core challenges of applying LLMs to Operations Research through a three-stage pipeline, as shown in Figure 2. We begin by establishing a robust data foundation. Firstly, we construct a high-quality seed dataset in Section 3.1.1 to mitigate data noise and inconsistencies. Next, we build the OR-ProcessQA dataset in Section 3.1.2, which provides the first process-supervised data in the OR domain with fine-grained, step-level annotations. Finally, we develop the Process Reward Model for OR domain (OR-PRM) in Section 3.3. This specialized PRM offers natural language critiques and corrections beyond scalar scores for OR reasoning steps. Our approach significantly enhances the reliability and performance of LLMs in OR by providing detailed, interpretable feedback throughout the solution process.

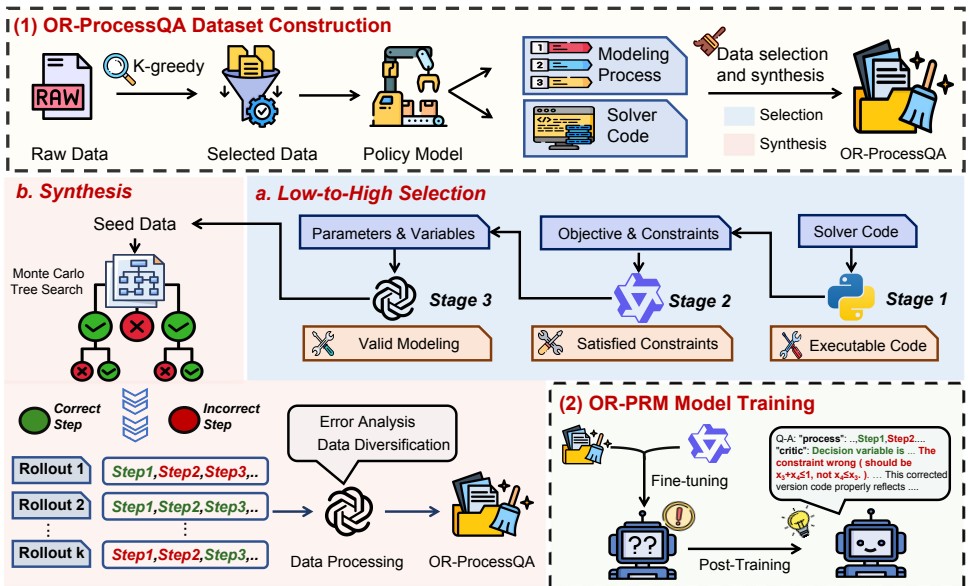

Figure 2: **Overview of our automated framework.** We first construct OR-ProcessQA through a three-stage filtering pipeline and MCTS-based trajectory generation with step-level verification. Built on this dataset, OR-PRM is trained to provide structured, stepwise feedback.

## 3.1 DATASET CONSTRUCTION

A high-quality dataset is essential to ensure the effectiveness of PRM supervision. We propose a stricter way to build the dataset. Specifically, we first create a cleaner seed dataset by careful filtering and many rounds of checking in Section 3.1.1. Then, we utilize this curated seed dataset to generate diverse and accurate process-annotated data in Section 3.1.2

### 3.1.1 SEED DATA CONSTRUCTION

In this section, we first standardize the problem representation for consistent generation. We then employ an existing strong OR model, **LLMOPT** Jiang et al. (2025), for solver code generation. Finally, we adopt a multi-stage procedure to filter out high-quality data.

**Problem representation**. We adopt LLMOPT as a generative policy that first produces each problem in the canonical five-element tuple form $(\mathcal{S}, \boldsymbol{\theta}, \boldsymbol{x}, f(\boldsymbol{x}), \boldsymbol{g}(\boldsymbol{x}) \leq \boldsymbol{c})$, ensuring compatibility with downstream validation and modeling stages. This policy-based generation ensures a mathematically well-formed and solver-agnostic structure from the start.

To enable consistent modeling and automated validation, we represent each problem $p$ via a compact five-element tuple:

$$p = \big(\mathcal{S},\ \boldsymbol{\theta},\ \boldsymbol{x},\ f(\boldsymbol{x}),\ \boldsymbol{g}(\boldsymbol{x}) \leq \boldsymbol{c}\big),$$

where $\mathcal{S}$ (index sets), $\boldsymbol{\theta}$ (parameters), $\boldsymbol{x}$ (variables), $f(\boldsymbol{x})$ (objective), and $\boldsymbol{g}(\boldsymbol{x}) \leq \boldsymbol{c}$ (constraints) collectively define the optimization task in canonical form $\min_{\boldsymbol{x}} f(\boldsymbol{x})$ s.t. $\boldsymbol{g}(\boldsymbol{x}) \leq \boldsymbol{c}$. This schema ensures solver-agnostic structure, enabling deterministic code-output validation against declared constraints and objectives, which is critical for scalable, error-free seed dataset construction.

**Solver Generation**. We directly use LLMOPT to auto-generate solver code tailored for each problem tuple, linking the mathematical formulation directly to an executable implementation.

**Multi-Stage Validation**. Each generated sample is then subjected to a three-stage validation pipeline to ensure high-quality reasoning. Samples were evaluated along three axes: code execution, constraint satisfaction, and modeling accuracy, and were discarded if they failed any stage.

1. *Code Execution:* We execute the provided code and verify that it runs without error and produces the expected output. This validates the code's executability and establishes the

resulting numerical solution $\hat{x}$ as the ground truth for subsequent constraint satisfaction checks.

2. *Constraint Satisfaction:* We employ Qwen3-8B Yang et al. (2025a) as a reasoning verifier: given the constraint expressions $g(x) \leq c$ from the five-element tuple and the numerical solution $\hat{x}$ produced by the solver code, it performs symbolic or numeric substitution to verify whether all constraints are satisfied. This enables automated, model-grounded feasibility checking without requiring additional code generation.

3. *Modeling Accuracy:* Finally, we use GPT-4o to validate whether the mathematical formulation accurately reflects the original problem statement. This ensures the five-element tuple $(\mathcal{S}, \theta, x, f, g)$ faithfully captures the problem semantics.

A sample is retained if and only if it passes all three validation stages: successful code execution, constraint satisfaction, and modeling accuracy. This integrated, generative process gave us a clean, reliable seed dataset.

### 3.1.2 STEP-WISE ANNOTATION GENERATION

Seed data can only support SFT but not PRM training, so we further expand it into step-wise trajectories and annotate them, obtaining a high-quality dataset suitable for PRM supervision. Specifically, this process consists of three parts: (1) automated step generation via MCTS based on the seed problems; (2) structured evaluation of each step using GPT-4o to identify potential errors; and (3) consistency filtering between MCTS and GPT-4o outputs to retain only logically sound trajectories.

**Automated Annotation via MCTS.** Following OmegaPRM Luo et al. (2024), we apply MCTS to problems from our seed dataset to sample solution trajectories. Correct steps are labeled 1.0, while the first error in any failed path is labeled 0.0. This process yields a raw dataset of over 550K annotated steps.

**Structured Error Analysis with GPT-4o.** To enhance reliability, we employ GPT-4o to systematically re-evaluate every candidate reasoning step. The model inspects each component in a predefined sequence: (1) parameter definitions, (2) objectives and constraints, (3) generated code, and (4) code execution output. Upon detecting the first error, it halts further analysis and outputs four structured fields:

- Issue: A natural language description of the error;
- Judgement: A binary label Correct or Incorrect;
- Corrected Version: The fixed content of the erroneous component;
- Corrected Step: The complete, revised reasoning step incorporating the fix.

This structured analysis ensures consistent, interpretable, and actionable feedback for training and refinement.

**Consensus-based Filtering.** We employ a dual-validation mechanism to curate the final training set. A sample is retained only if $\text{Label}_{\text{MCTS}}(s) = \text{Label}_{\text{GPT-4o}}(s)$, where Label denotes the binary validity label (correct or incorrect) and $s$ is the reasoning step.

Through this pipeline, we obtain high-confidence annotated samples, which constitute our final dataset: OR-Process-QA. This dataset strikes a balance between scale and precision, effectively supporting OR-PRMs fine-grained reward modeling and step-wise error correction capabilities.

### 3.2 GENERATIVE PRM FOR OR PROBLEM

**Traditional PRMs** often output a scalar score to represent the judgment. They employ a step-wise evaluation method. First, a scalar score is assigned to each reasoning step in a response. These scores are then aggregated, through methods like a weighted sum or by taking the minimum value, to calculate the final reward. However, traditional PRMs typically assign only a scalar value per step. This is not enough for complex tasks like operations research.

Such tasks require detailed analysis of variable relationships (e.g., $x$ over $\mathcal{S}$), constraint satisfaction ($g(x) \leq c$), and logical structure of the objective $f(x)$. Furthermore, while finding problems like syntax errors in code generation depends on the generation abilities of large language models, a

simple score is not enough to properly catch these potential issues especially when the code must align with the canonical form $\min_{\boldsymbol{x}} f(\boldsymbol{x})$ s.t. $\boldsymbol{g}(\boldsymbol{x}) \leq \boldsymbol{c}$.

**Generative PRM** replaces binary labels such as correct or incorrect with natural language judgments. During inference, the model generates a textual critique and judgment for each reasoning step, enabling interpretable and step-by-step evaluation. Inspired by GM-PRM Zhang et al. (2025a), we adopt a generative process reward modeling approach tailored for operations research tasks. Instead of assigning scalar scores to reasoning steps, our model generates natural language critiques and judgments for each component of the solution. This enables fine-grained, interpretable evaluation grounded in domain-specific logic.

Concretely, given an optimization problem $p = (\mathcal{S}, \boldsymbol{\theta}, \boldsymbol{x}, f(\boldsymbol{x}), \boldsymbol{g}(\boldsymbol{x}) \leq \boldsymbol{c})$ and its step-by-step solution, the model analyzes four key components in sequence: (1) variable definitions ($\boldsymbol{x}$ over $\mathcal{S}$, parameterized by $\boldsymbol{\theta}$), (2) objective $f(\boldsymbol{x})$ and constraints $\boldsymbol{g}(\boldsymbol{x}) \leq \boldsymbol{c}$, (3) code implementation (if present), and (4) final output. For each, it produces a brief intent statement, a focused analysis of critical issues, and a binary judgment correct or incorrect. If any component is judged incorrect, the model outputs a corrected version of the first flawed section only.

### 3.3 Training Objective

Our training process is structured in two main stages, to leverage our OR-ProcessQA dataset effectively. We first use Supervised Finetuning (SFT) to teach the model the fundamental format of generating critiques, followed by an Alignment phase with Direct Preference Optimization (DPO) to refine its logical judgment.

#### 3.3.1 Supervised Finetuning

The first stage, SFT, adapts a base model to the generative PRM task. The primary goal of SFT is to teach the model the correct format, style, and step-by-step reasoning process required for OR problem-solving.

Specifically, the model is trained on our high-quality annotated samples using a standard autoregressive next-token prediction objective. The input consists of a problem description and a candidate solution, while the target is the complete, structured critique generated during our data annotation pipeline (Section 3.3.2). The SFT loss function, $\mathcal{L}_{\text{SFT}}$, is defined as:

$$\mathcal{L}_{\text{SFT}}(\theta) = -\mathbb{E}_{(x,y)\sim\mathcal{D}_{\text{SFT}}} \left[ \sum_{t=1}^{T} \log P_\theta(y_t|x, y_{<t}) \right] \tag{1}$$

where $y$ represents the target sequence containing the four structured fields: **Issue**, **Judgement**, **Corrected Version**, and **Corrected Step**. This process teaches the model to perform the fine-grained, step-wise error analysis and correction that defines our generative PRM.

#### 3.3.2 Alignment

Supervised fine-tuning results in correctly formatted steps but lacks logical reliability. This is because the model simply imitates examples without deeper understanding. To address this, we use an alignment phase. This phase employs DPO to promote true logical reasoning.

**Direct Preference Optimization** We leverage our **OR-ProcessQA** dataset in conjunction with outputs from the SFT model: we re-run inference using the SFT model, identify failure cases (i.e., where the model produces incorrect or inferior reasoning), and construct preference pairs $(x, y_w, y_l)$ accordingly. For each prompt $x$, $y_w$ is the correct or superior reasoning step, while $y_l$ is the flawed step generated by the SFT model.

DPO directly optimizes the language model policy, $\pi_\theta$, to increase the likelihood of the preferred responses over the dispreferred ones, relative to a reference policy, $\pi_{\text{ref}}$. The DPO loss function is given by:

$$\mathcal{L}_{\text{DPO}}(\pi_\theta; \pi_{\text{ref}}) = -\mathbb{E}_{(x,y_w,y_l)\sim\mathcal{D}} \left[ \log \sigma \left( \beta \log \frac{\pi_\theta(y_w|x)}{\pi_{\text{ref}}(y_w|x)} - \beta \log \frac{\pi_\theta(y_l|x)}{\pi_{\text{ref}}(y_l|x)} \right) \right] \tag{2}$$

where $\beta$ is a temperature parameter controlling the strength of the preference, and $\sigma(\cdot)$ is the logistic function. This loss aligns the model with correct reasoning without requiring a separate reward model.

# 4 EXPERIMENTS AND ANALYSIS

In this section, we introduce our experimental setup for OR-PRM in Section 4.1. We then assess its performance in two distinct settings, discussed in Section 4.2. Finally, we present ablation studies in Section 4.3.

## 4.1 EXPERIMENTAL SETUP

**Model.** We evaluated the performance of OR-PRM when applied to several leading language models, including the Qwen2.5 series (7B, 14B, and 32B) and LLMOPT Jiang et al. (2025), a specialized model tailored for Operations Research. We chose Qwen2.5 because it offers a complete range of model sizes, enabling us to study scaling effects, and because it has demonstrated strong reasoning capabilities and wide adoption in recent LLM research.

**Benchmark.** We evaluated the model performance on a set of optimization benchmarks. However, even benchmarks in Operation Research contain serious errors Xiao et al. (2025); Jiang et al. (2025). To provide fair evaluation and preventing misleading answer, we utilized cleaned benchmarks from Xiao et al. (2025) to ensure the reliability of our results including Industry OR Huang et al. (2025a), Easy-LP Huang et al. (2025b), Complex-LP Huang et al. (2025b), NL4LP AhmadiTeshnizi et al. (2024), NL4OPT Ramamonjison et al. (2022).

**Training Details** To train OR-PRM, we use Qwen2.5-7B-Coder as base model. The training process was conducted in two stages on eight Nvidia A100 GPUs using DeepSpeed ZeRO-2 and bfloat16 precision. First, we perform supervised finetuning with a learning rate of 2e-5. Following this, the model undergoes Direct Preference Optimization (DPO) with a learning rate of 4e-5 and a beta of 0.2. A per-device batch size of 2 is applied in both training stages.

**Inference Details.** We evaluate OR-PRM under two complementary inference settings. The first focuses on *selection*, where multiple candidate reasoning paths are generated and OR-PRM identifies the most reliable one (**Best-of-N sampling**). The second emphasizes *refinement*, where OR-PRM critiques intermediate steps and guides the model toward improved solutions (**ModelingCritiqueGeneration pipeline**). For evaluation, correctness is verified numerically, and because many problems admit multiple solution paths, we compare only the final optimal value when reporting performance.

BEST-OF-N SAMPLING. By default, we set $N$=8. The model generates $N$ distinct Chain-of-Thought (CoT) Wei et al. (2022) reasoning paths with temperature 1.0. OR-PRM evaluates each reasoning step in every path as correct or incorrect, and selects the path containing the highest number of correct steps, favoring the most coherent and accurate reasoning trajectory.

MODELING, CRITIQUE, AND CODE GENERATION PIPELINE. In this setting, the base language model follows a structured three-stage workflow, guided by OR-PRM. First, the model constructs a formal problem modeling with step-by-step reasoning. Next, OR-PRM critiques each reasoning step by identifying potential errors or inconsistencies. Finally, the original modeling and its critique are concatenated and fed back into the model to guide the generation of executable Python code that satisfies predefined input-output specifications. This process enforces a self-correcting, implementation-aware reasoning trajectory through iterative feedback.

To thoroughly assess the efficacy of our proposed pipeline, we employed two primary evaluation metrics: **pass@1**, which measures the first-attempt correctness and reflects the model's immediate problem-solving capability; and **pass@8**, which evaluates the upper-bound potential when the model is allowed up to eight attempts, thereby revealing its capacity for self-correction and iterative refinement within a given search space.

| Model | IndustryOR | Easy-LP | Complex-LP | NL4LP | NL4OPT | ReSocratic | Overall |
|---|---|---|---|---|---|---|---|
| *Proprietary Models* | | | | | | | |
| GPT-4o | 40.5 | 69.5 | 35.1 | 56.2 | 53.1 | 47.9 | 50.4 |
| Deepseek-v3 | 66.7 | 91.9 | 39.6 | 92.7 | 76.5 | 73.9 | 73.6 |
| *Open-source Models* | | | | | | | |
| Qwen-2.5-7B | 19.0 | 49.7 | 12.6 | 50.0 | 41.3 | 36.7 | 34.9 |
| +PRM | 23.8 | 61.8 | 16.2 | 56.7 | 52.1 | 46.7 | 42.9 |
| | **+4.8** | **+12.1** | **+3.6** | **+6.7** | **+10.8** | **+10.0** | **+8.0** |
| Qwen-2.5-14B | 35.7 | 66.2 | 3.6 | 75.8 | 61.0 | 50.4 | 48.8 |
| +PRM | 45.2 | 89.4 | 12.6 | 86.5 | 67.6 | 66.7 | 61.3 |
| | **+9.5** | **+23.2** | **+9.0** | **+10.7** | **+6.6** | **+16.3** | **+12.5** |
| Qwen-2.5-32B | 47.6 | 80.0 | 8.2 | 87.1 | 68.5 | 66.3 | 59.6 |
| +PRM | 57.1 | 96.0 | 32.4 | 89.3 | 74.2 | 72.7 | 70.3 |
| | **+9.5** | **+16.0** | **+24.2** | **+2.2** | **+5.7** | **+6.4** | **+10.7** |
| LLM-OPT | 52.4 | 96.0 | 48.6 | 90.4 | 81.7 | 72.2 | 73.6 |
| +PRM | 59.5 | 97.8 | 67.6 | 93.8 | 85.0 | 79.2 | 80.5 |
| | **+7.1** | **+1.8** | **+19.0** | **+3.4** | **+3.3** | **+7.0** | **+6.9** |

Table 1: **Results on Six Reasoning Benchmarks.** Experimental results demonstrate that using OR-PRM as the critic model significantly enhances reasoning performance under the Best-of-8 evaluation strategy. The line in blue indicates performance improvement.

## 4.2 MAIN RESULTS

**Best-of-N Sampling.** As shown in Table 1, OR-PRM consistently and significantly enhances reasoning performance across different scales of the Qwen model family. It achieves uniform gains on the Qwen2.5 Yang et al. (2024) series (7B32B) and the specialized model LLMOPT Jiang et al. (2025), **demonstrating its effectiveness and strong scalability with respect to model size**. Notably, on the 14B model, OR-PRM achieves the *highest average improvement of nearly 12.5%*.

Moreover, the performance gains introduced by OR-PRM are consistently evident across tasks of varying difficulty levels. On the most challenging Complex-LP benchmark, Qwen2.5-32B attains an impressive absolute improvement of 24.2%. For relatively easier benchmarks such as Easy-LP, the 14B model achieves substantial gains of 23.2%. Even for LLMOPT, a model already extensively optimized for reasoning and exhibiting strong performance on difficult tasks, OR-PRM contributes an additional 19.0% improvement on Complex-LP. These results further *substantiate the effectiveness of OR-PRM in accurately identifying and prioritizing high-quality reasoning steps under demanding conditions*.

**Results of Modeling-Critique-Code Pipeline.** As shown in Figure 3, OR-PRM consistently demonstrates remarkable performance enhancements across both the prominent open-source model Qwen-2.5-14B and the advanced closed-source model GPT-4o.

The most substantial improvements are particularly evident on the challenging Complex-LP benchmark, underscoring potent ability of OR-PRM to tackle intricate problems. The pass@1 accuracy for Qwen2.5-14B surged by an impressive 23.4%, while even the state-of-the-art GPT-4o achieved a notable increase of 8.1%. The gains in pass@8 are also notable: Qwen2.5-14B witnessed a significant rise of 36.1%, and GPT-4o improved by 6.3%.

These gains underscore ability of OR-PRM to raise the reasoning ceiling by effectively recovering correct solutions from initial failures. Even when the first attempt falters, OR-PRM enables iterative correction, enhancing robustness under uncertainty and complexity. On the simpler Easy-LP benchmark, it still yields consistent 24% improvements, demonstrating reliability across task difficulty.

At the heart of OR-PRM is its critic componentan intelligent feedback loop that evaluates each reasoning step. It reinforces correct steps and precisely diagnoses errors, offering targeted guidance rather than binary judgments. This fine-grained feedback helps the model iteratively refine its reasoning, much like a human learner, leading to notable accuracy gains. Such interactive error correction is key to broad effectiveness of OR-PRM across models and tasks.

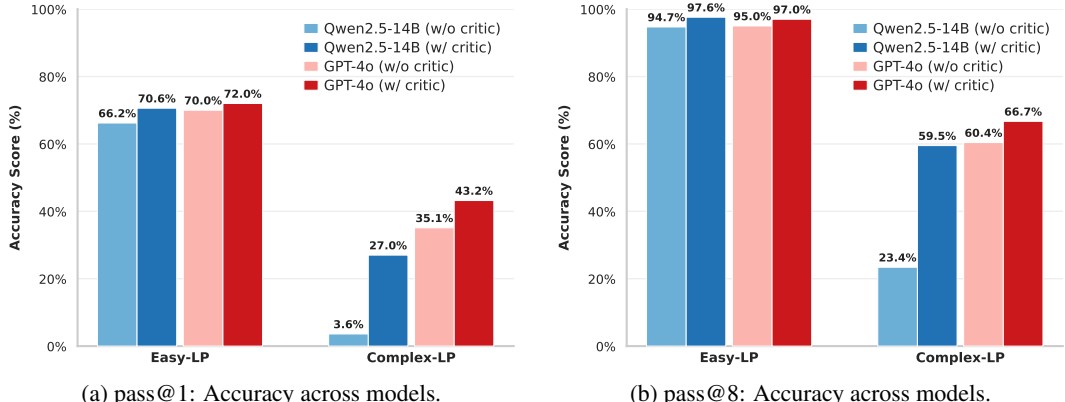

(a) pass@1: Accuracy across models.  (b) pass@8: Accuracy across models.

Figure 3: **OR-PRM enhances optimization ability across models.** It consistently improves performance on both open-source (Qwen2.5-14B) and closed-source (GPT-4o) models, and enables solving problems that remain unsolved even with 8 samples.

## 4.3 ABLATION STUDIES

In this section, we analyze the effectiveness of model alignment via DPO and examine performance trends across task difficulty levels. The results are presented in Table 2.

| Method | Easy-LP | Complex-LP | Average |
|---|---|---|---|
| Pass@8 | 94.7% | 23.4% | 59.1% |
| self-consistency | 50.8% | 3.6% | 27.2% |
| **OR-PRM (Ours)** | **89.4**% | **12.6%** | **51.0%** |
| OR-PRM (SFT) | 79.6% | 6.3% | 43.0% |
| Qwen2.5 (Zero shot) | 72.1% | 9.9% | 41.0% |
| self-consistency (filtered null) | 88.3% | 9.9% | 49.6% |

Table 2: **Ablation results.** Results on Qwen2.5-14B.

**Effectiveness of Model Alignment**   Our ablation study confirms the effectiveness of Direct Preference Optimization (DPO) within the OR-PRM model training. As shown in Table 2, the full model incorporating DPO on top of SFT achieves an average accuracy of 51.0%. This represents an 8.0% absolute improvement over the SFT-only baseline (43.0%), demonstrating DPO's crucial role in improving model. Other baselines include the Qwen2.5 (Zero shot) model, which represents the raw base model performance without any SFT or DPO training, and the self-consistency (filtered null) approach, which performs majority voting on the $N = 8$ paths after filtering out those that fail to produce a valid numerical objective value.

**Performance Across Task Difficulty Levels**   As shown in Table 2, OR-PRM consistently outperforms the Major Voting baseline across both easy and challenging benchmarks. This performance demonstrates that OR-PRM has the ability to detect a significant majority of errors within reasoning paths across both easy and challenging benchmarks.

## 4.4 DISCUSSION

We further discuss the limitations in current training data and fine-grained discrimination capability, with future directions outlined below.

Our OR-PRM performs well on the new OR-ProcessQA dataset. However, it is hard to provide a comparison, as existing datasets cannot be used for PRM training. Furthermore, our Best-of-N performance is strong, but it still falls short of the theoretical upper bound. This performance gap is

mainly attributed to the current size of our dataset and model. Therefore, we will expand the training data in the future, to make the model better at detecting subtle reasoning errors.

## 5 CONCLUSION AND LIMITATION

In this work, we introduce OR-PRM, the first Process Reward Model (PRM) tailored for Operations Research (OR), designed to address the core challenge of reliable LLM reasoning in this domain. Our investigation revealed that the primary obstacle to developing such a model was the pervasive unreliability of existing OR datasets, which prevents PRMs from learning to accurately distinguish between valid and invalid reasoning steps. To overcome this fundamental data bottleneck, we first curated a high-quality seed dataset and expanded it into OR-ProcessQA, the first OR dataset with reliable, step-level correctness annotations. This provided the essential foundation for our model. Building on this unique resource, OR-PRM delivers structured, step-level feedback rather than a single scalar score. Experiments demonstrate that our approach is highly effective. OR-PRM substantially improves LLM performance, yielding an average 12.5% gain in the Best-of-N setting and notable robustness when serving as a critic during inference. These results underscore the value of process-oriented supervision for LLM reasoning in OR, suggesting a promising direction for developing more trustworthy AI in other domains that require verifiable, step-by-step logic. Indeed, these successful results affirm the foundational value of our dataset. However, we also acknowledge a current limitation: the lack of datasets to compare. Therefore, to enhance the credibility of our research findings and support broader applications, we plan to further expand and refine our dataset, including by increasing the diversity of problem types and solver environments.

## ACKNOWLEDGMENT

This work was supported in part by the National Natural Science Foundation of China (Grant No. 6250074347), and the Major Key Project of PCL under Grants PCL2025AS10 and PCL2024A06.

## ETHICS STATEMENT

This work focuses on improving the reliability of large language models (LLMs) in Operations Research (OR) through process-oriented supervision. No human subjects were directly involved in data collection. Our dataset, OR-ProcessQA, is derived entirely from synthetic sources and existing public benchmarks, followed by automated filtering and GPT-4o verification. All data are anonymized, contain no personal or sensitive information, and comply with open licensing terms of the source datasets.

Potential risks include the possibility of misuse of OR-capable LLMs in high-stakes decision making (e.g., logistics, finance, or defense). To mitigate such risks, our method emphasizes correctness, transparency, and logical consistency, making model outputs more interpretable and auditable. We also release detailed dataset construction protocols to encourage responsible use.

We declare that there are no conflicts of interest or external sponsorship that might unduly influence the presented results. This research adheres to the ICLR Code of Ethics.

## REPRODUCIBILITY STATEMENT

We have made extensive efforts to ensure reproducibility.

- **Dataset:** The construction pipeline for the high-quality seed dataset and OR-ProcessQA is fully described in Section 3.2, with additional filtering rules and statistics detailed in the Appendix.

- **Models:** The architecture and training procedure of OR-PRM are explained in Section 3.3, with hyperparameters, optimization details, and ablation results provided in the supplementary materials.

- **Code & Resources:** We will release anonymized source code, dataset filtering scripts, and training configurations as supplementary material.

- **Evaluation:** All metrics, baselines, and Best-of-N setups are documented in Section 4 and Appendix.

These resources, combined with detailed documentation, ensure that independent researchers can reproduce the reported results.

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

## A    THE USE OF LARGE LANGUAGE MODELS (LLMS)

Large Language Models were employed as general-purpose assistive tools throughout the research process. Specifically, LLMs were used to aid and polish the writing of this manuscript, including refining grammar, improving clarity, and restructuring sentences for better readability.

In this work, LLMs were utilized for data processing. Specifically, GPT-4o was used to assess the modeling accuracy of the initial data and to perform step-by-step error analysis and annotation of the process. Meanwhile, Qwen3-8B served as a reasoning verifier, automatically checking constraint satisfaction via numeric substitution for feasibility validation. All LLM-generated content underwent cross-validation or manual spot-checking to ensure the models functioned strictly as assistive tools.

All outputs generated by LLMs were critically evaluated and edited by the authors, and no content was used without verification. The use of LLMs did not replace human intellectual contributions but served to accelerate and enhance various stages of the research workflow.

## B    BENCHMARKS AND EVALUATION

We conduct experiments on the following real-world optimization task datasets.

- **IndustryOR** Huang et al. (2025a) is the first industrial-grade dataset specifically designed for optimization modeling. It integrates real-world operations research (OR) problems from eight different industries, covering five types of optimization problemslinear programming, integer programming, mixed-integer programming, nonlinear programming, and other special problem typesacross three difficulty levels. The training set contains 3,000 instances without optimal solutions, while the test set includes 100 instances with optimal solutions, aiming to comprehensively evaluate a model's ability to solve optimization problems in real-world industrial scenarios.

- **MAMO** Li et al. (2025) offers a novel optimization dataset for evaluating the mathematical modeling capabilities of large language models. The dataset is divided into two parts: **Easy LP**, which contains 652 high school-level Mixed-Integer Linear Programming (MILP) problems for foundational learning, and **Complex LP**, which provides 211 undergraduate-level challenges that blend complex applications of linear and mixed-integer linear programming. Notably, this dataset does not include any Nonlinear Programming (NLP) problems.

- **NLP4LP** AhmadiTeshnizi et al. (2024) dataset features 65 curated cases from optimization textbooks and lecture notes. These cases cover various application areas, including facility location, network flow, scheduling, and portfolio management. Each instance includes a detailed problem description, a parameter data file, and the optimal value derived from textbook solutions or manual solving, offering a range of complex optimization challenges of varying difficulty.

- **NL4OPT** Ramamonjison et al. (2022) is a curated dataset developed from the competition of the same name, which focuses on converting natural language descriptions of optimization problems into solver-ready code. The dataset primarily addresses Linear Programming (LP) problems across different scenarios but lacks more complex Mixed-Integer Programming and Scheduling (MIPS) problems. In experiments, a filtered test set of 213 high-quality instances was used.

- **ReSocratic** Yang et al. (2025b) is an innovative reverse data synthesis method that generates high-quality operations research optimization problems by following a unique from answer to question path. Starting with 27 well-designed seed demonstrations, this method uses the DeepSeek-V2 model to progressively generate new structured cases, ensuring quality through a dual-filter mechanism. Finally, it reverse-translates these formatted cases into natural language problems and corresponding executable code, ultimately creating the RESOCRATIC-29K dataset.

We use the clean version from Xiao et al. (2025), an accurate subset of the benchmark. Specifically, we employ Qwen2.5-14B-Instruct to extract the corresponding optimal values and then compare them with the ground truth.

# C  SEED DATASET

## C.1  SAMPLING STATISTICS OF THE EXISTING DATASET

| Dataset | Sampling Size | Error Rate |
|---|---|---|
| Opt-Math-train | 500 | $\geq 16\%$ |
| IndustryOR-train | 500 | $\geq 31\%$ |
| Resocratic-train | 500 | $\geq 30\%$ |
| Evo-step | 500 | $\geq 25\%$ |

Table 3: Sample data from different synthetic datasets.

Table 3 shows the error rates across several datasets. We also performed an error attribution analysis on Industry-OR and found that approximately 84% of errors were modeling errors (e.g., missing constraints, incorrect objective functions, or unit mismatches), 11% were code implementation errors (e.g., variable definition or logic mistakes), and only about 4% were result inconsistencies (i.e., output solutions violating constraints or not matching computed values).

## C.2  DETAILS OF BUILD SEED DATASET

**Code Execution**   We perform a straightforward execution of the generated code and then evaluate two criteria: (1) whether the execution completes successfully without errors, and (2) whether the output matches the ground truth.

**Constraint Satisfaction**   In this stage, we use an Qwen3-8B verifier to confirm the feasibility of the solver's numerical solution. The verifier is given the mathematical constraints and the solution, and it performs symbolic or numeric substitution to automatically check if all conditions are met, as demonstrated in the manufacturing example (Figure 4).

**Modeling Accuracy**   This final and most critical stage employs a powerful LLM to evaluate if the mathematical formulation faithfully captures the intent of the original problem statement. It identifies crucial semantic flaws, such as a misaligned objective function (e.g., maximizing total parts instead of complete sets). This check ensures the model is not just feasible but also semantically correct, as illustrated in the factory production example (Figure 5).

## C.3  FINAL SEED DATASET

| Dataset | Size | Full Size |
|---|---|---|
| Opt-Math-train | 3282 | 210000 |
| IndustryOR-train | 1375 | 3000 |
| Resocratic-train | 4036 | 29000 |
| Evo-step | 3351 | 4464 |

Table 4: Sample data from different Synthetic.

We sampled data from four sources: Opt-Math Lu et al. (2025), IndustryOR Huang et al. (2025a), Resocratic Yang et al. (2025b), and Evo-step Wu et al. (2025). For the Opt-Math and Resocratic datasets, we first applied k-greedy filtering to the initial data. Following a three-stage filtering process and deduplication, we obtained a final dataset of 8,656 instances. We manually checked 100 samples from the final data, and the accuracy is approximately 96%.

---

**Example: Verifying Constraint Satisfaction**

**Question**: A manufacturing company produces five electronic devices: Smartphones, Tablets, Laptops, Smartwatches, and Cameras. The profit per unit and labor hours required are given in the table below:

| Device | Profit ($) | Labor Hours |
|--------|-----------|-------------|
| Smartphones | 100 | 5 |
| Tablets | 150 | 8 |
| Laptops | 200 | 10 |
| Smartwatches | 50 | 3 |
| Cameras | 300 | 12 |

The objective is to maximize total profit.

**Solution** The optimization solver returns the candidate solution:

$$\hat{x} = (x_1, x_2, x_3, x_4, x_5) = (0,\ 500,\ 200,\ 133,\ 300),$$

**Feasibility Verification by Qwen3-8B**

corresponding to (Smartphones, Tablets, Laptops, Smartwatches, Cameras).
Qwen3-8B substitutes $\hat{x}$ into each constraint expression to verify feasibility:

- **Labor hours**: $5(0) + 8(500) + 10(200) + 3(133) + 12(300) = 9999 \leq 10000$ ✓
- **Smartphones + Tablets**: $0 + 500 = 500 \leq 500$         ✓
- **Laptops**: $200 \leq 200$         ✓
- **Smartwatches**: $133 \geq 100$         ✓
- **Cameras**: $300 \leq 300$         ✓

All constraints are satisfied, confirming that $\hat{x}$ is a feasible solution.

Figure 4: Constraint Satisfaction Example

**Problem:** A company has three factories (X, Y, Z) that produce three different components (1, 2, 3) required for a product. Each factory has a limited number of production hours, and their productivity (units/hour) for each component varies. The objective is to allocate production hours across the factories to maximize the number of complete sets of components that can be assembled.

| Factory | Hours | Product. 1 | Product. 2 | Product. 3 |
|---------|-------|-----------|-----------|-----------|
| X | 120 | 14 | 11 | 9 |
| Y | 140 | 16 | 10 | 7 |
| Z | 100 | 12 | 15 | 8 |

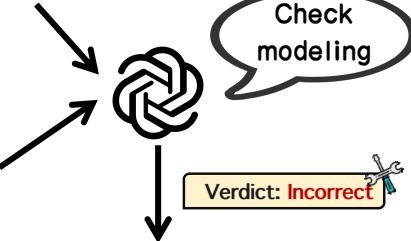

Check modeling

Verdict: **Incorrect**

##Sets:
  Factories F = {X, Y, Z}
  Components C = {1, 2, 3}
##Parameters:
  Productivity: p_X1=14, p_X2=11, ...
  Max hours: H_X=120, H_Y=140, H_Z=100
##Variables:
  x_X1, x_X2, x_X3: Hours allocated in Factory X f
  or components 1, 2, 3....
##Objective Function:
  Maximize (14*x_X1 + 11*x_X2 + 9*x_X3 + 16*x_Y1 + ...)
  (Maximizing the **total quantity** of all individual components)
##Constraints:
  x_X1 + x_X2 + x_X3 ≤ 120 (Production hour limit for Factory X),
  ......,All x variables ≥ 0 (Non-negativity)

Modeling Process

**Analysis:** The formulation maximizes the total number of units produced, but the problem description requires maximizing the number of completed sets of components. This implies a need for balance among the components produced, which is not addressed in the current objective function. The formulation lacks constraints or an objective that ensures the production of complete sets, such as a additional constraints to balance production of different components.

Figure 5: Modeling Accuracy Example: LLM checks whether the modeling aligns with the intended meaning of the question.

# D OR-PROCESSQA CONSTRUCTION

## D.1 DETAILS OF MONTE CARLO TREE SEARCH

Monte Carlo Tree Search is a heuristic search algorithm for sequential decision-making in large state spaces. In our OR-PRM, we employ it as the first stage of our automated data synthesis pipeline to efficiently generate a large volume of candidate reasoning steps along with their preliminary correctness labels. MCTS iteratively constructs a search tree $T = (V, E)$, where each node $v \in V$ represents a partial solution (i.e., a reasoning prefix), and each edge $(v, a) \in E$ represents a reasoning step $a$ generated by the policy model.

**Selection**  Starting from the root node (i.e., the original problem), the algorithm recursively selects child nodes to balance exploitation and exploration. It adopt the following Upper Confidence Bound applied to Trees formula.

$$a^* = \arg\max_{a \in A(v)} \left[ Q(v, a) + c \cdot \sqrt{\frac{\ln N(v)}{N(v, a)}} \right] \tag{3}$$

Here, $Q(v, a)$ is the average probability of reaching the correct final answer after taking action $a$ from node $v$; $N(v)$ and $N(v, a)$ are the visit counts for node $v$ and edge $(v, a)$, respectively; $c$ is a constant controlling the strength of exploration.

**Expansion**  When the search reaches a leaf node $v_l$ that still has unexplored actions, the algorithm invokes the policy model to generate a new reasoning step $a$ based on the current state $v_l$, thereby creating a new node $v_{new}$ and adding it to the tree.

**Simulation**  From the newly expanded node $v_{new}$, the algorithm performs one or more rollout simulations by prompting the policy model to autoregressively generate a complete reasoning path to a final answer. The simulation outcome $z$ is a binary reward: $z = 1$ if the final answer is correct, otherwise $z = 0$.

**Backpropagation**  The simulation result $z$ is propagated back up the search path, updating the statistics for all traversed nodes:

$$N(v) \leftarrow N(v) + 1 \tag{4}$$

$$Q(v, a) \leftarrow Q(v, a) + \frac{z - Q(v, a)}{N(v, a)} \tag{5}$$

In the OR-PRM data synthesis pipeline, the core value of MCTS lies in its automation. We configured key hyperparameters to balance exploration diversity and efficiency: sampling temperature T = 1.0 , Top-k sampling k = 50 , nucleus sampling (Top-p) p = 0.9 , and exploration coefficient c = 1.0. Through this structured search, the algorithm efficiently generates over 550,000 candidate reasoning steps with preliminary labels from our carefully curated set of 8,000 seed problems. This provides ample raw material for the subsequent stages: structured error analysis and consensus-based filtering performed by GPT-4o. The preliminary hard labels (0 or 1) generated by MCTS, combined with the detailed natural language critiques from GPT-4o, ultimately produce the high-quality, high-reliability OR-ProcessQA dataset, forming a solid foundation for training the powerful generative process reward model, OR-PRM.

## D.2 STRUCTURED ERROR ANALYSIS WITH GPT-4O

To generate the rich, step-level supervision required for our generative PRM, we employ GPT-4o as an expert annotator.

We initially sampled approximately 40,000 data points to form the starting dataset. This final consensus-based filtering step proved to be highly selective, discarding approximately 50% of the candidate annotations and resulting in a high-fidelity dataset of roughly 20,000 samples.

We inspected a sample of 100 data points and found that the critics judgments were correct in 95% of the cases.

## E  INFERENCE WITH OR-PRM

OR-PRM can identify different scenarios, such as correct cases, modeling errors, code errors, etc. Three examples as follow:

**MODELING**

**Problem Statement:**

The Li family plans to invest their retirement fund in commercial real estate. Property 1 has an annual income of \$12,500, Property 2 has an annual income of \$35,000, Property 3 has an annual income of \$23,000, and Property 4 has an annual income of \$100,000. The decision to be made is whether to buy or not buy each property, not the quantity, as there is only one property per property. Help them decide which properties to purchase to maximize their annual income. Property 1 costs \$1.5 million, Property 2 costs \$2.1 million, Property 3 costs \$2.3 million, and Property 4 costs \$4.2 million. The Li family's budget is \$7 million. If they purchase Property 4, then they cannot purchase Property 3.

## MODELING

**SETS**

$$\mathcal{P} = \{1, 2, 3, 4\}$$

**PARAMETERS**

$$c_i : \mathcal{P} \to \mathbb{R}, \quad c_i = \begin{cases} 12500 & \text{if } i = 1 \\ 35000 & \text{if } i = 2 \\ 23000 & \text{if } i = 3 \\ 100000 & \text{if } i = 4 \end{cases}$$

$$p_i : \mathcal{P} \to \mathbb{R}, \quad p_i = \begin{cases} 150000 & \text{if } i = 1 \\ 210000 & \text{if } i = 2 \\ 230000 & \text{if } i = 3 \\ 420000 & \text{if } i = 4 \end{cases}$$

$$B \in \mathbb{R}, \quad B = 7000000$$

**VARIABLES**

$$x_i \in \{0, 1\}, \quad i \in \mathcal{P}$$

**OBJECTIVE**

$$\max_{\boldsymbol{x} \in \mathcal{X}} \sum_{i=1}^{4} c_i x_i$$

**CONSTRAINTS**

$$\sum_{i=1}^{4} p_i x_i \leq B$$

$$x_3 + x_4 \leq 1$$

$$\boldsymbol{x} = (x_1, x_2, x_3, x_4)^\top$$

## PYOMO IMPLEMENTATION

```
from pyomo.environ import *
# --- Model Initialization ---
model = ConcreteModel()
# ... Sets (P) and Parameters (c, p, B) are defined here based on the model.
# --- Key Modeling Logic ---
# 1. Define Variables
# x_i is a binary variable, 1 if property i is purchased, 0 otherwise.
model.x = Var(model.P, domain=Binary)
# 2. Define Objective Function
# Maximize the total annual income from purchased properties.
model.income = Objective(
    expr=sum(model.c[i] * model.x[i] for i in model.P),
    sense=maximize
)
# 3. Define Constraints
# The total cost of purchased properties cannot exceed the budget B.
model.budget_constraint = Constraint(
    expr=sum(model.p[i] * model.x[i] for i in model.P) <= model.B
)
# If property 4 is purchased, property 3 cannot be.
model.exclusion_constraint = Constraint(
    expr=model.x[3] + model.x[4] <= 1
)
# ... Solver is called and solution is processed here.
```

## ANSWER

**Objective value**: 135000.0
**Optimal Solution**: $x_1 = 0.0$, $x_2 = 1.0$, $x_3 = 0.0$, $x_4 = 1.0$

Figure 6: **Correct Case:** Modeling Process.

**STRUCTURED ERROR ANALYSIS**

**1. Variable Definition Analysis**
Intent: Define decision variables.
Analysis: The decision variable $x_i$ is defined correctly as binary, indicating whether to purchase each property.
Judgement: Correct

- - - - - - - - - - - - - - - - - - - - - - - - - - - - - - - - - - - - - - - - - - - - - - - - - - -

**2. Objective & Constraint Analysis**
Intent: Formulate model.
Objective: Correct. The objective maximizes total annual income from selected properties.
Constraints: - Budget constraint ($\sum_{i=1}^{4} p_i x_i \leq B$) is correct. - Constraint on purchasing properties 3 and 4 ($x_3 + x_4 \leq 1$) is correct.
Judgement: Correct

- - - - - - - - - - - - - - - - - - - - - - - - - - - - - - - - - - - - - - - - - - - - - - - - - - -

**3. Code Analysis**
Intent: Implement model in Pyomo/Python.
Analysis: - Sets, parameters, and variables are defined correctly in the code. - The objective function and constraints match those specified in the problem statement. - The output matches the expected result based on the problem description.
Judgement: Correct

- - - - - - - - - - - - - - - - - - - - - - - - - - - - - - - - - - - - - - - - - - - - - - - - - - -

**4. Final Answer Analysis**
Intent: Report solution or error.
Analysis: The reported solution shows that properties 2 and 4 should be purchased, yielding an optimal annual income of \$135,000. This aligns with the problem requirements and constraints.
Judgement: Correct

- - - - - - - - - - - - - - - - - - - - - - - - - - - - - - - - - - - - - - - - - - - - - - - - - - -

No corrections needed; all parts of the solution are accurate.

Figure 7: **Correct Case:** Structured Error Analysis.

---

**MODELING**

## SOLUTION

### SETS

$\mathcal{P}$ represents the set of properties, with $\mathcal{P} = \{1, 2, 3, 4\}$

### VARIABLES

$x_i \in \{0, 1\}$, $i \in \mathcal{P}$, where $x_i = 1$ if property $i$ is purchased, $x_i = 0$ otherwise.

### PARAMETERS

$p_i$ is the annual income of property $i$, $i \in \mathcal{P}$, $p_1 = 12500$, $p_2 = 35000$, $p_3 = 23000$, $p_4 = 100000$
$c_i$ is the cost of property $i$, $i \in \mathcal{P}$, $c_1 = 1500000$, $c_2 = 2100000$, $c_3 = 2300000$, $c_4 = 4200000$
b is the budget, b = 7000000

### OBJECTIVE

$$\max_{\boldsymbol{x} \in \{0,1\}^4} \sum_{i \in \mathcal{P}} p_i x_i$$

### CONSTRAINTS

$\sum_{i \in \mathcal{P}} c_i x_i \leq b$, $\boldsymbol{x} \in \{0, 1\}^4$

---

### PYOMO IMPLEMENTATION

```python
from pyomo.environ import *

# --- Model Initialization ---
model = ConcreteModel()

# ... Sets (P) and Parameters (p, c, budget) are defined here based on the model.

# --- Key Modeling Logic ---

# 1. Define Variables
# x_i is a binary variable, 1 if property i is purchased, 0 otherwise.
model.x = Var(model.P, domain=Binary)

# 2. Define Objective Function
# Maximize the total annual income from purchased properties.
model.objective = Objective(
    expr=sum(model.p[i] * model.x[i] for i in model.P),
    sense=maximize
)

# 3. Define Constraints
# The total cost of purchased properties cannot exceed the budget.
model.budget_constraint = Constraint(
    expr=sum(model.c[i] * self.model.x[i] for i in self.model.P) <= self.model.budget
)

# <<<< INCORRECT CONSTRAINT >>>>
# The problem states: "If property 4 is purchased, then property 3 cannot be."
# The correct logic is x4 + x3 <= 1.
# The code below incorrectly implements this as x4 <= x3.
model.order_constraint = Constraint(
    expr=model.x[4] <= model.x[3]
)

# ... Solver is called and solution is processed here.
```

---

### ANSWER

**Objective value**: 123000.0
**Optimal Solution**: $x_1 = 0.0$, $x_2 = 0.0$, $x_3 = 1.0$, $x_4 = 1.0$

Figure 8: **Incorrect Answer Case:** Modeling Process.

---

**STRUCTURED ERROR ANALYSIS**

**1. Variable Definition Analysis**
Intent: Define decision variables.
Analysis: The decision variable $x_i$ is defined correctly as binary, indicating whether each property should be purchased.
Judgement: Correct

- - - - - - - - - - - - - - - - - - - - - - - - - - - - - - - - - - - - - - - - - - - - - -

**2. Objective & Constraint Analysis**
Intent: Formulate model.
Objective: Correct. The objective maximizes the total annual income from selected properties.
Constraints: - Budget constraint is correct; it ensures that the sum of costs does not exceed the budget. - Order constraint is incorrect. It states $x_4 \leq x_3$, meaning if property 4 is purchased, then property 3 must also be purchased. However, the original problem specifies that if property 4 is purchased, property 3 cannot be purchased.
Judgement: Incorrect due to the constraint error.

- - - - - - - - - - - - - - - - - - - - - - - - - - - - - - - - - - - - - - - - - - - - - -

**3. Code Analysis**
Intent: Implement model in Pyomo/Python.
Analysis: - Binary decision variables $x_i$ are correctly implemented. - The budget constraint is implemented correctly. - There is an implementation error in the order constraint. Instead of $x_4 \leq x_3$, it should be $x_4 + x_3 \leq 1$.
Judgement: Incorrect

- - - - - - - - - - - - - - - - - - - - - - - - - - - - - - - - - - - - - - - - - - - - - -

**4. Final Answer Analysis**
Intent: Report solution or error.
Analysis: The reported solution suggests purchasing properties 3 and 4, yielding a total annual income of 123,000 dollars. This contradicts the problem statement since it implies purchasing both properties 3 and 4 simultaneously, violating the logical condition specified.
Judgement: Incorrect

- - - - - - - - - - - - - - - - - - - - - - - - - - - - - - - - - - - - - - - - - - - - - -

**5. Corrected Step: Code Implementation**

```
# ... (imports and model setup are identical to the incorrect version)

def define_constraints(self):
    # The budget constraint remains the same.
    self.model.budget_constraint = Constraint(
        expr=sum(self.model.c[i] * self.model.x[i] for i in self.model.P)
        <= self.model.budget
    )

    # CORRECTED LOGIC for the exclusion constraint:
    # This now correctly implements the rule "If property 4 is purchased,
    # property 3 cannot be" as x4 + x3 <= 1.
    self.model.order_constraint = Constraint(
        expr=self.model.x[4] + self.model.x[3] <= 1
    )

# ... (objective definition and solver calls remain the same)
```

- - - - - - - - - - - - - - - - - - - - - - - - - - - - - - - - - - - - - - - - - - - - - -

This corrected version properly reflects the logical constraint that if property 4 is purchased, property 3 cannot be purchased.

Figure 9: **Incorrect Answer Case:** Structured Error Analysis.

---

**MODELING**

## SOLUTION

### SETS

$$\text{Properties} = \{1, 2, 3, 4\}$$

### PARAMETERS

Income(Properties) = (12500, 35000, 23000, 100000),
Cost(Properties) = (1500000, 2100000, 2300000, 4200000)

### VARIABLES

$$\text{Purchase(Properties)} = \begin{cases} 1 & \text{if Property } i \text{ is purchased,} \\ 0 & \text{otherwise} \end{cases}$$

### OBJECTIVE

$$\max_{\textbf{Purchase}} \sum_{i \in \text{Properties}} \text{Income}(i) \cdot \text{Purchase}(i)$$

### CONSTRAINTS

$$\sum_{i \in \text{Properties}} \text{Cost}(i) \cdot \text{Purchase}(i) \leq 7000000$$

$$\text{Purchase}(3) \leq 1 - \text{Purchase}(4)$$

---

## PYOMO IMPLEMENTATION

```python
from pyomo.environ import *

# --- Model Initialization ---
model = ConcreteModel()
# ... Sets, Parameters, and Variables are defined here.
# ... Objective function is also defined.

# --- Key Modeling Logic (Constraints) ---

# Correctly implemented budget constraint
def budget_constraint_rule(model):
    return sum(model.Cost[i] * model.Purchase[i] for i in model.Properties) <= 7000000
model.BudgetConstraint = Constraint(rule=budget_constraint_rule)

# <<<< INCORRECT CONSTRAINT IMPLEMENTATION >>>>
# The constraint Purchase(3) <= 1 - Purchase(4) is a single, scalar constraint.
# It should not be indexed over the set of all properties. The code below
# incorrectly defines it as an indexed constraint, leading to a parameter error.
def exclusivity_constraint_rule(model, prop3, prop4): #<-- Incorrect function signature
    # This rule expects two indices (prop3, prop4) but Pyomo only provides one.
    if prop3 == 3 and prop4 == 4:
        return model.Purchase[3] <= 1 - model.Purchase[4]
    else:
        return Constraint.Skip

# This line causes the error because the rule's signature does not match
# how Pyomo calls it for an indexed constraint.
model.ExclusivityConstraint = Constraint(model.Properties, rule=exclusivity_constraint_rule)

# ... Solver call leads to a TypeError.
```

---

## ANSWER

Figure 10: **Code Execution Error Case:** Modeling Process.

---

**STRUCTURED ERROR ANALYSIS**

**1. Variable Definition Analysis**
Intent: Define decision variables.
Analysis: The variable 'Purchase' is defined correctly as binary, indicating whether each property should be purchased or not.
Judgement: Correct

- - - - - - - - - - - - - - - - - - - - - - - - - - - - - - - - - - - - - - - - - - -

**2. Objective & Constraint Analysis**
Intent: Formulate model.
Objective: Correct. The objective maximizes total income based on selected properties.
Constraints: - Budget constraint is correct; it ensures that the total cost does not exceed $7 million. - Exclusivity constraint is intended to prevent purchasing both properties 3 and 4 simultaneously. However, the implementation has an issue due to the way constraints are indexed and formulated.
Judgement: Incorrect (due to implementation issue in exclusivity constraint)

- - - - - - - - - - - - - - - - - - - - - - - - - - - - - - - - - - - - - - - - - - -

**3. Code Analysis**
Intent: Implement model in Pyomo/Python.
Analysis: There is a critical error in defining the exclusivity constraint. The function exclusivity constraint rule expects two arguments (prop3, prop4) but only receives one (prop3). This mismatch causes a TypeError. Additionally, the constraint logic needs adjustment to properly enforce mutual exclusion between properties 3 and 4.
Judgement: Incorrect

- - - - - - - - - - - - - - - - - - - - - - - - - - - - - - - - - - - - - - - - - - -

**4. Final Answer Analysis**
Intent: Report solution or error.
Analysis: Since the code fails at the implementation level, there is no valid output or solution reported. The error message indicates a failure in constructing the exclusivity constraint.
Judgement: Incorrect

- - - - - - - - - - - - - - - - - - - - - - - - - - - - - - - - - - - - - - - - - - -

**5. Corrected Step: Code Implementation**

```
# ... (imports and model setup are identical to the previous version)

def define_constraints(self):
    # The budget constraint is a scalar constraint and remains the same.
    self.model.budget_constraint = Constraint(
        expr=sum(self.model.c[i] * self.model.x[i] for i in self.model.P)
        <= self.model.budget
    )

    # CORRECTED IMPLEMENTATION for the exclusion constraint:
    # The previous version incorrectly tried to define this as an indexed
    # constraint over a set. The correct approach is to define it as a
    # single, scalar constraint, as it does not depend on any index.
    self.model.order_constraint = Constraint(
        expr=self.model.x[4] + self.model.x[3] <= 1
    )

# ... (objective definition and solver calls remain the same)
```

- - - - - - - - - - - - - - - - - - - - - - - - - - - - - - - - - - - - - - - - - - -

This corrected version defines the exclusivity constraint more clearly and appropriately, ensuring that properties 3 and 4 cannot both be purchased.

Figure 11: **Code Execution Error Case:** Structured Error Analysis.

## F   PROMPTS USED IN EXPERIMENTS

---

**CRITIC PROMPT**

```
\label{critic}
You are an expert in Operations Research (OR).

You will be given an optimization problem and (optionally) a step-
    by-step solution, which may or may not include code.

Task: Review the solution. Analyze each applicable part in order.
    Be concise only highlight critical errors or omissions. Skip
    any section if the input doesn't contain it (e.g., no code skip
     Code Analysis).

Evaluate in this order:

1. Variable Definitions
2. Objective Function and Constraints
3. Code Implementation (if provided)
4. Final Answer / Output

Question:
{Question}

Solution Steps:
{Solution}

Output Format (be brief and precise):

1. Variable Definition Analysis
- Intent: [e.g., Define decision variables]
- Analysis: [Only note missing, redundant, or misdefined variables]
- Judgement: [Correct/Incorrect]

2. Objective and Constraint Analysis
- Intent: [e.g., Formulate model]
- Objective: [Correct? Brief reason if wrong]
- Constraints: [Missing/incorrect? List only key issues]
- Judgement: [Correct/Incorrect]

3. Code Analysis (Skip if no code)
- Intent: Implement model in Pyomo/Python
- Analysis: [Only flag mismatches: missing vars/constraints, wrong
    indexing, type errors]
- Judgement: [Correct/Incorrect or Skipped]

4. Final Answer Analysis
- Intent: [e.g., Report solution or error]
- Analysis: [Must show valid optimal solution AND objective value.
    If output contains ANY error/traceback (e.g., SyntaxError,
    AttributeError)  Incorrect. [Plausible? Error meaningful? Root
    cause if wrong]]
- Judgement: [Correct/Incorrect]

Corrected Step (Only if any part above is Incorrect)
- [Rewrite only the first incorrect section  e.g., fix constraints
    or variables  in full, clearly labeled.]
```

**QUESTION TO MODELING PROMPT**

```
You are an expert in Operations Research (OR).
The following is an optimization problem. You need to write the
    corresponding Pyomo code based on the problem description and
    information provided.

The problem description is as follows:
```
{ques}
```

The following is the five-element model of an optimization problem:
```
{five}
```

Please write the corresponding Pyomo code. Please add 'from pyomo.
    environ import *' at the beginning of your code (You can add
    other 'import' as well). Please print the optimal solution and
    the value of the objective function. Please do not output the
    running log. You need to write it in the form of a class and
    add a main function:

```python
[write your code here]
```
```

**MODELING TO CODE PROMPT**

```
You are an expert in Operations Research (OR).
The five-element model is the abstraction of an optimization
    problem, which transforms specific problem scenarios into
    formal mathematical problems. You need to write the
    corresponding Pyomo code based on the five-element model
    provided.

The following is the five-element model of an optimization problem:
```
{five}
```

Please write the corresponding Pyomo code. Please add 'from pyomo.
    environ import *' at the beginning of your code (You can add
    other 'import' as well). Please print the optimal solution and
    the value of the objective function. Please do not output the
    running log. You need to write it in the form of a class and
    add a main function:

```python
[write your code here]
```
```

**EXTRACT ANSWER PROMPT**

```
You are an expert in Operations Research (OR).
Your task is to precisely extract and return exactly one line from
    the multi-line text provided below. This line must state the
    final optimization value (e.g., maximum profit, minimum cost,
    or total objective value).

    ## Core Instructions

    - **Exact Extraction**: The returned content must be a complete
    , unmodified line as it appears in the original text.
    - **Single Output**: Your response must contain only the
    extracted line. Do not add any prefixes, suffixes, explanations
    , introductory phrases, or extra formatting.
    - **Keyword Recognition**: Prioritize identifying and
    extracting the line that contains common optimization keywords
    such as:
    - `cost`
    - `profit`
    - `objective`
    - `value`
    - `revenue`
    - `optimal value`
    - 'Total'

    Text to analyze:
    ---
    {text}
```

