# OpenReview forum: "OR-PRM: A Process Reward Model for Algorithmic Problem in Operations Research"
_ICLR.cc/2026/Conference — ICLR 2026 Poster_

### Official Review · Reviewer_mUPb · 2025-10-20

**Soundness:** 3
**Presentation:** 3
**Contribution:** 3
**Rating:** 8
**Confidence:** 3

**Summary:**

This paper introduces OR-PRM, the first Process Reward Model (PRM) tailored for Operations Research (OR) reasoning tasks. The work targets a fundamental challenge: large language models (LLMs) often fail to produce reliable, logically consistent reasoning in OR problems involving optimization modeling, constraints, and solver code generation.

To address this, the authors propose a three-stage pipeline:

Data construction – Build a high-quality, process-annotated dataset (OR-ProcessQA) through careful seed curation, constraint validation, and semantic verification.

Monte Carlo Tree Search (MCTS) – Generate diverse reasoning trajectories and preliminary correctness labels automatically.

Process Reward Model (OR-PRM) – Train a generative PRM using Supervised Fine-Tuning (SFT) followed by Direct Preference Optimization (DPO), enabling fine-grained, interpretable step-level feedback for OR reasoning.

Extensive experiments on multiple open-source (Qwen2.5, LLMOPT) and closed-source (GPT-4o) models show that OR-PRM substantially improves reasoning accuracy, robustness, and interpretability.

**Strengths:**

Well-motivated and novel contribution: Addresses a clear gap between generic reasoning PRMs and domain-specific reasoning in mathematical optimization. OR-PRM is the first reward model explicitly designed to evaluate reasoning steps in mathematical optimization, going beyond scalar scoring to produce structured, natural-language critiques and corrections.

Methodologically rigorous: Multi-stage pipeline (SFT + DPO) is logically constructed, with explicit checks (execution, constraints, semantics) to ensure data correctness.

Excellent technical clarity: The paper defines each component precisely (seed data, MCTS, PRM training), making it reproducible and interpretable.

Comprehensive evaluation: Benchmarks include both open- and closed-source models, with ablation studies isolating the effect of DPO alignment.

Strong empirical results and interpretability: The improvement is not just numerical but also interpretable, showing how OR-PRM critiques and corrects reasoning steps.

**Weaknesses:**

It is not clear to me whether the evaluation pipeline can serve as the ground truth. Since LLMOPT and GPT-4o may make mistakes. It is unclear how far the evaluation is from having an OR expert evaluate the reasoning steps.

**Questions:**

The pipeline heavily depends on LLMOPT. It would be nice to discuss if a different model were used to generate the data, how much impact that would have on the results.

It seems that the OR problems have data that appeared in the problem description in scalar form. What if the data is stored in CSV files? Would the pipeline still work?

---

> ### Author Response · Authors · 2025-11-23
>
> Thank you for raising these important points. We address them as follows:
>
> W1. It is not clear to me whether the evaluation pipeline can serve as the ground truth
> > Thank you for your concern about our data quality. We manually checked 100 ground-truth answers and found 96% were correct, showing our evaluation pipeline works well. We will include the relevant details in the paper.
>
> Q1. The pipeline heavily depends on LLMOPT
> > Thank you for this important question regarding our model selection.
> > We chose LLMOPT primarily for two reasons: (1) it is a strong OR-specific model that produces more valid reasoning paths, and (2) it generates diverse reasoning trajectories for training. We also briefly evaluated alternative models. While these alternatives can also yield high-quality reasoning paths, they present practical limitations:
>
> >    1. Using a strong model through it API (Deepseek) would significantly increase annotation costs, especially API expenses;
> >    2. Using a larger open-source model (Qwen2.5-72B) would incur higher time and computational overhead;
>
> >  Thus, both of the above models can generate satisfactory paths, and using either of the above models is also feasible, selecting LLMOPT is largely a trade-off—striking a good balance between data quality, diversity, and practical feasibility.
>
> Q2. If the data is stored in CSV files? Would the pipeline still work?
> > Yes, our pipeline fully supports CSV files—we designed it with this scenario in mind from the start.

---

### Official Review · Reviewer_xTZk · 2025-10-27

**Soundness:** 3
**Presentation:** 3
**Contribution:** 3
**Rating:** 4
**Confidence:** 3

**Summary:**

The paper introduces OR-PRM, a domain-specialized Process Reward Model for Operations Research (OR). The authors (1) diagnose high noise in existing OR datasets; (2) curate a cleaner seed set via a three-stage pipeline; (3) generate step-wise trajectories with MCTS; (4) have GPT-4o perform structured, step-level judgments and corrections; (5) train a generative PRM that outputs critiques rather than scalar scores; and (6) show gains under Best-of-N selection and a Modeling→Critique→Code pipeline, reporting up to +12.5 pp average improvement across benchmarks and model sizes (Qwen2.5 series, LLMOPT).

**Strengths:**

1. PRMs are a natural fit because OR solutions require step-wise logical validity (not only final objective values). The paper targets this gap explicitly.

2. Three-stage filtering for a seed set, MCTS exploration, and GPT-4o step audits with issue/judgement/correction fields—this is more structured than typical PRM pipelines in math reasoning.

3. Returning natural-language critiques and a corrected first error goes beyond scalar PRMs and mirrors trends in “corrective” PRMs in broader reasoning literature.

4. Best-of-N and model-agnostic critic usage show uniform gains; the Complex-LP results are especially notable (e.g., +24.2 pp on Qwen2.5-32B).

**Weaknesses:**

1. The paper should deeply contrast with OmegaPRM (automated process supervision via MCTS) which established scalable step-labeling at large scale in math reasoning (1.5M annotations) (https://arxiv.org/abs/2406.06592), and with recent generative/critic-style PRMs designed to explain and correct steps such as GM-PRM and VisualPRM/Athena-PRM (multimodal, but methodologically very close in “PRM generates critiques + refines BoN”) (https://arxiv.org/abs/2508.04088). The proposed “generative PRM” feels incremental w.r.t. these trends.

2. The paper claims >30% severe flaws in mainstream OR data and a rigorous three-stage filter, but I don’t see inter-rater reliability, error taxonomy distributions, or random spot-check protocols beyond illustrative examples. Given PRM sensitivity to label noise (documented in PRM surveys/lessons) (https://arxiv.org/abs/2501.07301) , stronger auditing is necessary.

3. Step labeling and final verification heavily rely on strong LLMs (GPT-4o/Qwen verifiers) inside the pipeline that also guide critique at inference. Without cross-model, cross-vendor checks and held-out validators, there is potential circularity (the critic agrees with the validator it was trained/selected with).

4. Since BoN/selection tends to give large lifts, the paper should compare against strong non-PRM baselines (self-consistency; majority vote with route-length filters; retrieval-augmented PRM or OOD-robust PRMs) that recent works show to be competitive, e.g., Retrieval-Augmented PRM (R-PRM / RAPRM) and R-PRM variants focusing on OOD and data bootstrapping. (https://arxiv.org/abs/2502.14361) Current ablations (e.g., “Major Voting”) look weak and not representative of the SOTA toolkit.

5. OR is a heterogeneous space (LP/MILP/NLP/CP-SAT with industry quirks). I don’t see OOD splits (new templates, new constraint families, solver switches) or robustness to noisy/problematic instances (e.g., infeasible but realistic specs). OOD weaknesses are a known pain point for PRMs. (https://arxiv.org/abs/2502.14361)

6. The Modeling→Critique→Code pipeline may conflate several effects: (i) data curation, (ii) structured prompts, (iii) critic guidance. A factorized ablation (swap in a scalar PRM; blind the critic to code; remove “Corrected Step”) would clarify whether “generative PRM” itself is the key.

7.Important but under-specified: exact prompts, MCTS hyper-parameters per task, GPT-4o temperature and refusal handling, and the policy/critic decoupling at test time (who conditions on whom).

**Questions:**

1. How often does GPT-4o disagree with the MCTS label? Show confusion matrices and human audits on a random 500-step sample. How many “corrections” by GPT-4o were later proven wrong?

2. If you allow the critic to emit corrections (you already do), can you implement Refined-BoN like GM-PRM and report deltas? (https://arxiv.org/abs/2508.04088)

3. Evaluate with new solver backends (e.g., Pyomo→PuLP / OR-Tools), new template families, and noisy text to test OOD per RAPRM concerns. (https://arxiv.org/abs/2502.14361)

4. Ablate the “generative” aspect: Replace the critic with (i) scalar PRM, (ii) critique-only (no correction), (iii) correction-only (no explanation). Which component drives Complex-LP gains?

5. Add self-consistency, vote-with-length-normalization, and retrieval-augmented PRM. Current “Major Voting” is too weak to claim SOTA.(https://arxiv.org/abs/2502.14361)

6. End-to-end token/cost for data curation + training + inference (critic calls per step)? Compare with OmegaPRM’s cost per step label. (https://arxiv.org/abs/2406.06592)

7. Failure modes: Provide qualitative cases where OR-PRM gives confident but wrong constraints/objectives (classic OR pitfall), and whether the system catches feasible but semantically wrong models.

8. For 100 randomly sampled instances, have expert OR graders rate the usefulness and correctness of critiques (Likert & error-type taxonomy). Your current evidence relies mostly on automatic verifiers.

9.  As far as I know, the algorithm complexity of MCTS is relatively high, and I am interested in the impact of introducing MCTS on computational efficiency.

10. If we don't use GPT-4o, which is a relatively good model, but some relatively poor models, can your method still demonstrate robustness?

---

> ### Author Response · Authors · 2025-11-27
>
> Thank you for your feedback, we’ve responded in detail below and welcome any further questions.
>
> W1.  Relation to OmegaPRM and GM-PRM:
> > We acknowledge that OR-PRM builds upon established paradigms, OmegaPRM and GM-PRM. Our work represents the systematic effort to extend these methods to the operations research (OR) domain. The core contribution lies in adapting these frameworks to OR-specific challenges, such as data noise and the duality of textual and code-based reasoning, and demonstrating significant performance gains over standard baselines.
>
> W2. Data Quality, Auditing and Error Taxonomy  & W3 & Q1 & Q8 :
>
> >Your concern about data quality is well-founded. To address this, Our audit of 100 randomly sampled instances confirmed that approximately 96% of the ground-truth labels are correct.
>
> >Regarding potential circularity (W3): This is a critical point. While we selected GPT-4o for its state-of-the-art capabilities, we recognize that independent validation is essential. Our manual audit serves as this independent check. We found that the Critic's correctness judgments were accurate in approximately 95% of cases, confirming reliable evaluation.
>
> >Noise in Code Corrections: Conversely, we observed approximately 23% noise in the generated "Corrected Step," primarily within the code implementation segments. This stems from the fact that our current correction synthesis relies on LLM generation without real-time solver execution feedback. This finding identifies a clear path for future work: we plan to introduce stronger supervision by integrating solver execution feedback during the correction generation phase, ensuring that the corrected code is as rigorous as the modeling logic.
>
> W4. Baselines (Major Voting is weak; R-PRM).
> >
> >  We apologize for the unclear terminology; our "Major Voting" baseline is the standard implementation of Self-Consistency (Wang et al., 2022), and we will correct the term in the revision. As shown in Table 2, our OR-PRM significantly outperforms this strong baseline.
> >
>
> >  And we completely agree that R-PRM is a very important and clever paper. It uses a unique way of retrieving information to help the model generalize better and solve out-of-distribution (OOD) problems.
> > Due to time constraints, we will strive to reproduce R-PRM’s experiments in our future work. For the revised version, we will add a detailed discussion of its key contributions to the “Related Work” section. A direct comparison remains our top priority.
>
> W5. OOD Robustness and Heterogeneity
>
> > We fully agree that OOD generalization and heterogeneity are critical challenges, as highlighted by RetrievalPRM (Zhu et al., 2025). Our current dataset primarily prioritizes accuracy, with expansion to diverse solvers planned as a subsequent step. In our revision, we will explicitly acknowledge this limitation and discuss how retrieval-augmented methods serve as a key solution for enhancing OOD robustness.
>
> W6. Factorized Ablation (Generative vs Scalar):
> > Thank you for this insightful question. You have correctly identified a key challenge in disentangling the individual contributions within our pipeline. We agree that a fully factorized ablation would be ideal, and we address your specific points as follows:
> >
> >    1. We did explore a scalar PRM, but it proved to be ineffective. The training data from our Monte Carlo sampling has a highly imbalanced label distribution, which caused the scalar model to learn a trivial policy of habitually classifying most steps as incorrect. This provided little useful signal for genuine refinement.
> >    2. We also acknowledge that our data collection process inherently couples the natural language critique with the Corrected Step,which makes a fully isolated ablation of these two components non-trivial at this stage.
> >    3. However, we have strong evidence that the generative guidance itself is the critical driver of performance. Our full pipeline experiment achieves a Pass@8 of 23.4% on the challenging Complex-LP benchmark. The Pass@8 metric measures the upper bound of a model's problem-solving capability through self-correction. A scalar PRM can only select from existing solutions; it cannot guide the model to fix a failed attempt. The substantial boost in Pass@8 is direct proof that our generative PRM enables iterative self-correction, thereby raising the model's performance ceiling in a way a scalar model cannot.
>
> W7. Missing Details:
> > Thank you for pointing this out. We will include all the specific details as required in the appendix.

---

> ### Author Response · Authors · 2025-11-27
>
> Q6 & Q9. End-to-end token/cost for data curation + training + inference (critic calls per step)? Compare with OmegaPRM’s cost per step label.
> > For the data synthesis phase (MCTS), we utilized 56 Nvidia V100 GPUs for approximately 4 days to generate the MCTS Dataset. This is a one-time offline cost. Crucially, MCTS is not used during inference; our method only requires a linear forward pass for the critic, ensuring high computational efficiency for deployment.
>
> Q7. Failure modes:
> > Catching Feasible butWrong Models:
> >
> >Yes. This is the core strength of OR-PRM.
> Qualitative Case: In the investment problem, the generated code runs without error (feasible), but incorrectly models  . OR-PRM correctly detects this and labels it Incorrect. We will include it in the appendix.
> >
> > Failure Mode (Confident but Wrong):
> >
> >The primary failure mode is over-correction. In cases with ambiguous problem descriptions, OR-PRM may confidently hallucinate that a constraint is missing when it is not actually required.
>
> Q10.  we don't use GPT-4o, which is a relatively good model, but some relatively poor models, can your method still demonstrate robustness?
> >  A strong model (GPT-4o) is indispensable for data synthesis. Weaker models fail to distinguish between syntactic correctness (code runs) and semantic faithfulness (logic is true), and they struggle to follow the complex, multi-step annotation format.

---

### Official Review · Reviewer_gX4v · 2025-10-31

**Soundness:** 2
**Presentation:** 2
**Contribution:** 3
**Rating:** 2
**Confidence:** 3

**Summary:**

The authors develop the first Process Reward Model (PRM), which is specifically intended for Operations Research. Based on prior research that existing synthetic datasets in this area are significantly flawed, the manuscript proposes a filtering pipeline to initially construct a "seed" dataset of problems and their description in a formalized way as well as a solution. Each problem is validated in multiple ways by ensuring that the reference code runs and yields the correct result, that constraints are valid as well as that the mathematical model to solve represents the problem faithfully. This dataset is then used to construct several reasoning trajectories of multiple reasoning steps based on Monte Carlo Tree Search (MCTS), where each step is validated, to generate a dataset for training (called OR-ProcessQA). A process reward model, OR-PRM, is then trained on this dataset to provide dense rewards for guiding the reasoning effectively. The reward is not a single score, but instead provides feedback in natural and corrections in natural language, so is more like a critic model.

**Strengths:**

* new dataset seems to be addressing a need in this field, if existing datasets are so severely lacking
* writing seems to be mostly clear, especially the description of the dataset construction

**Weaknesses:**

* field of Operations Research is never properly introduced
* novelty is a bit diminished by over-claiming: abstract/introduction/conclusion give the impression that the analysis of the existing datasets is done by the authors of this manuscript, but judging from the related work section it was done by someone else
  * this directly hurts the motivation for the new dataset, since the problems in existing datasets are not discussed in detail
* dataset statistics are mostly missing, for example:
  * number of samples in the (seed) dataset
    * What is a sample?
  * average length of trajectories, i.e. number of steps
  * number of trajectories per sample
  * how many failed/successful trajectories
  * dataset composition from the original datasets (partially in Appendix C.2)
  * little empirical analysis of the dataset creation pipeline
* limited evaluation:
  * generation (not the primary focus of the manuscript): while different model sizes are used, they are only from one model family, leaving the question whether the approach generalizes to other model families
  * only a single model, small, trained as a reward model, so generalization is a big question
  * analysis focuses too much on the employed generative models and less on the capabilities of the reward model
  * no baselines for the first scenario (Best-of-N) besides the original model
  * unclear baselines for second scenario:
    * How does pass@1 with critic work? It considers the proposed corrected step/solution?
    * How does pass@8 without critic work?
  * potential baseline: Use of a general purpose model with the same prompt to act as an critic?
  * missing cost analysis

minor issues:
* it seems that the wrong cite command is used in general, so brackets around the references are missing
* related work:
  * title - seems to be not all caps like the other section titles
  * line 147: for some reason "offline" is incorrectly hyphenated
* Figure 2:
  * you might want to make the bubble (bottom row, roughly the middle) a bit bigger, so that "Data Diversification" does not touch the outline
  * same for the bubble inside "OR-PRM Model Training"
* titles of 3.1 and 3.1.1: You might want to properly capitalize the titles.
* 4.1:
  * line 327: "Specifically, Industry OR..." - not a proper sentence
  * line 331: "OR-PRM, We" - "we" should not be capitalized
  * line 346: reference for CoT?
* references:
  * cited differently than other arXiv references (can only be surmised from the URL):
    * [DeepSeek-AI 2024]
    * [Luo et al. 2024]
    * [Ma et al. 2024]
    * [Zhang et al. 2025a]
  * additionally consider capitalizing the titles to be consistent, especially abbreviations and proper names (which is sometimes done):
    * [Huang et al. 2025a/b]
    * [Ma et al. 2024]
    * [OpenAI 2025]
    * [Wang et al. 2025]
    * [Wu et al. 2025]
    * [Xiao et al. 2024]
    * [Xiao et al. 2025]
    * [Xie et al. 2023]
    * [Yang et al. 2025b]
    * [Zhang et al. 2025a]
    * [Zhou et al. 2025]
  * missing place of publication: [Wu et al. 2025]
  * [Xiao et al. 2025] was published IJCAI '25
* Appendix B:
  * NL4OPT: The number of samples (245) seems to differ from Table 3.
  * line 711: Wrong table reference (Table 3 instead of 4)?
* Figure 5, caption: "Example:LLM" - missing whitespace
* Appendix E, line 858: "etc.Three" - missing whitespace; "Three examples as follow:" - not a proper English sentence
* Appendix F: title sounds a bit strange

**Questions:**

* 3.1.1: What does $\hat x$ represent? The new solution? But if the expected output is already known, why not use that as ground truth?
* 3.1.2: What is $\mathcal{L}$? I suppose it is not the loss function?
* 3.2: How are Generative PRMs different from a general critic model employed for each reasoning step?
* Table 1: What would have been the improvements, if the best solution would have been selected based on the ground truth, i.e. what would have been the upper bound of improvement when generating additional trajectories?
  * This would give context how well OR-PRM selects reasoning chains or whether improvements "just" stem from the additional generation, i.e. more token use.
  * this sentence in 4.4 seems to suggest that such data exists: "our Best-of-N performance is strong, but it still falls short of the theoretical upper bound" (line 466)
  * Why was OR-PRM not used in this setting for the proprietary models
* Table 2:
  * What does Qwen2.5 (Zero Shot) do? Why are the results different than in Figure 3?
    * same for OR-PRM (Ours)
  * How does majority voting (filtered null) work?
  * Which scenario is used for the ablation study? probably the second one
* Did you try any out of distribution benchmarks?
* 4.4/conclusion: How would expanding the dataset help to create credible baselines?
* Appendix F, Critic Prompt: seems only for the full solution
  * Can we actually call this a Process Reward Model, if it does not look at individual reasoning steps?
  * What is difference between the two prompts on page 24? Where are they applied?

---

> ### Author Response · Authors · 2025-11-23
>
> Thank you for your thoughtful review questions! We’d be happy to discuss them, and please feel free to ask more.
>
> W1. Regarding Operations Research (OR) has not been adequately introduced
> > We agree this is a valid concern. To address it, we’ll add a concise paragraph in the Introduction outlining OR’s core ideas, its real-world applications (e.g., logistics, finance, scheduling), and why OR poses unique challenges for LLMS, ensuring all readers have the necessary context.
>
> W2. Overstatement weakens novelty and motivation.
> > Thank you for your valuable feedback. We will revise the section to better highlight our original contribution.
> >
> > We cite Xiao et al. (2025) only to contextualize general awareness of OR data quality issues. Our work is instead motivated by our own systematic audit, which found that at most over 30% of samples in major public OR datasets contain critical flaws. This finding directly drives our three-stage filter.
> >
> > To clarify, we will:
> > 1. Clearly separate general observations from our validation in the main text, emphasizing the latter as the basis of our method.
> > 2. Add an appendix with audit statistics.
>
> W3. Dataset statistics are mostly missing
> > Although we already include basic seed information and illustrative examples, we will further enhance the appendix by adding a dedicated table and accompanying text that comprehensively summarize the seed dataset and OR-ProcessQA.
>
> W4. On Limited Evaluation
> *   On Generalization
>     > Your concern is crucial. Our experiments demonstrate the generalization capability of OR-PRM in two aspects:
>     >
>     > 1.  Scalability within the same model family: OR-PRM consistently yields improvements across the Qwen2.5 series (7B/14B/32B) (see Table 1), validating its strong scaling ability.
>     > 2.  Effectiveness on state-of-the-art closed-source models: When used as an external critic to guide GPT-4o, OR-PRM still significantly boosts performance (see Figure 3), indicating it can enhance models of different capability levels across various architectures.
>
> *   On Using a Single Model
>     > We chose to train a 7B reward model by design: to balance specialized expertise in Operations Research (OR) with inference efficiency and cost. This aligns with the current trend of using smaller, specialized reward models to guide larger, general-purpose models.
>
> *   Analysis Focuses Too Much on Generative Models
>     > We believe the Best-of-N (BoN) experiments directly quantify the discriminative ability of the Reward Model. Given a fixed generative model, performance improvement depends entirely on the Reward Model's ability to accurately select the best solution from N candidates. OR-PRM achieves an average improvement of 12.5% on BoN tasks, and up to 24.2% on Complex-LP, which strongly demonstrates its capability to effectively distinguish between valid and invalid reasoning processes.
>
> *   No Baselines for Best-of-N
>     > The "Major Voting" in Table 2 serves as the standard Self-Consistency baseline (Wang et al., 2022). Our results show that OR-PRM's selection mechanism significantly outperforms this simple majority voting baseline, proving that process-based reward discrimination is more reliable than merely relying on consensus of final answers.
>
> *   Unclear Baselines for the Second Scenario
>     > We define the two settings as follows:
>     >
>     > *   Pass@8 without critic: This refers to the results from randomly sampling 8 solutions generated independently by the base model. It can be considered the upper bound of sampling without guidance.
>     > *   Pass@1 with critic: This is our core "correction pipeline." The model generates an initial formulation, OR-PRM provides a critique, and the model then revises its solution based on this feedback to produce the final result.
>
> *   Missing a Potential Baseline
>     > Table 2 shows that using a general-purpose, un-finetuned model (e.g., Qwen2.5-Instruct) as the critic yields significantly worse performance, demonstrating the necessity of domain-specific training. Here, "zero-shot" refers to directly using the un-finetuned Qwen2.5-Instruct model as the critic.
>     >
>     > Explanation of "Zero-shot": In this context, a "Zero-shot Critic" refers to using a general, instruction-tuned model to evaluate and correct OR problems without any specific training or fine-tuning on OR-related data. The results show that this kind of "layman" guidance is ineffective.
>
> *   Missing Cost Analysis
>     > *   Training Cost: The primary costs stem from MCTS exploration and GPT-4o API calls for data annotation. However, this is a one-time, offline investment.
>     > *   Inference Cost: We intentionally use a small 7B model as the critic, making its inference cost extremely low. Compared to simply scaling up the generative model (e.g., from 14B to 72B) to improve performance, introducing a lightweight OR-PRM for verification and correction is a highly cost-effective solution.

---

> ### Author Response · Authors · 2025-11-23
>
> Response to Minor Issues
> > Thank you again for your thorough review. We will address all noted formatting and typographical issues as follows:
> >
> > - Consistently use the correct `\cite{}` command to ensure citations appear with parentheses.
> > - Standardize the capitalization of all section headings for consistent style.
> > - Correct spelling and grammatical errors (e.g." capitalization in "OR-PRM, We," etc.).
> > - Adjust the layout of Figure 2 to prevent text from overlapping with the border.
> > - Fix the spacing issue in the caption of Figure 5 (e.g., "Example: LLM").
> > - Correct typographical errors in the appendix (e.g., "etc.Three" → "etc. Three").
> > - Standardize the formatting of all references, particularly the citation style for arXiv preprints and the capitalization of titles.
>
> Q1. What does $\hat{x}$ represent?
> > We apologize for the lack of clarity. $\hat{x}$ represents the numerical solution produced by executing the generated solver code. For instance, $\hat{x} = [10, 5]$ implies assigning 10 trucks to Route A and 5 to Route B. We will indicate this in the text.
>
> > Because this is the data construction phase, no pre-existing answer exists. As stated in Step 1 ("Code Execution"), the successful execution of the code "establishes $\hat{x}\$ as ground truth." This ground truth is then used in Step 2 ("Constraint Satisfaction") to verify if the code's logic is consistent with the problem's mathematical constraints.
>
> Q2. What is $L$?
> > We apologize for the confusion caused by this typographical error. In the expression $L_{\mathrm{MCTS}}(s) = L_{\mathrm{GPT-4o}}(s)$, the symbol $L$ denotes the **binary label** (i.e., 1 for "Correct", 0 for "Incorrect") assigned to a reasoning step $s$. This equation indicates that a sample is retained only when the label derived from MCTS matches the judgment of GPT-4o. We will correct this in Section 3.1.2 by updating the notation to $Label_{\mathrm{MCTS}}(s) = Label_{\mathrm{GPT-4o}}(s)$ to eliminate ambiguity.
>
> Q3. What distinguishes Generative PRMs from generic critic models?
> > This is an insightful question. We will add a clarifying paragraph in Section 3.2 to highlight the key differences:
> > - Domain Specialization: OR-PRM is trained on our carefully curated OR-ProcessQA dataset, enabling it to internalize domain-specific structures and knowledge inherent to operations research problems—such as nuances in constraints, variables, and objectives. These are details that generic models struggle to grasp via zero-shot prompting alone.
> > - Structured Feedback: OR-PRM is trained to generate structured feedback (including the problem statement, correctness judgment, a revised solution, and specific correction steps). This standardized output is more reliable than the free-form text typically produced by generic critics and can be directly integrated into an automated self-correction pipeline.
> > - Efficiency and Cost: Our OR-PRM (based on a 7B-parameter model) is significantly faster and more cost-effective at inference time compared to using large frontier models like GPT-4o as critics.
>
> Q4. Table 1:
> > 1.  Upper Bound Analysis
> >
> >     We appreciate the reviewer’s insightful suggestion regarding the "theoretical upper bound." The relevant results are reported in Table 2 (Ablation Study):
> >     The “Pass@8” row represents the Oracle upper bound—the probability that at least one of the 8 sampled trajectories is correct
> >    - On Easy-LP, OR-PRM (89.4%) approaches the upper bound (94.7%), indicating it reliably identifies correct solutions when present.
> >    - On Complex-LP, OR-PRM (12.6%) lags behind the upper bound (23.4%) but still significantly outperforms the majority voting baseline (3.6%), demonstrating its superior selection capability on harder tasks.
> > 2. Why Proprietary Models Were Not Included in Best-of-N
> >
> >    Due to high API costs, we did not include proprietary models like GPT-4o in the Best-of-N setting (Table 1). Instead, as shown in Figure 3, we adopted a more cost-effective “Modeling–Critique–Code” pipeline (1 generation + 1 critique), which better reflects real-world usage. Results confirm that OR-PRM effectively boosts GPT-4o’s performance, demonstrating its generalization to closed-source models.
> Q5. Table 2:
> > 1.  What does Qwen2.5 (Zero Shot) do? Why are the results different than in Figure 3?
> >     Same as in weakness. In this context, a "Zero-shot Critic" refers to using a general, instruction-tuned model to evaluate and correct OR problems without any specific training or fine-tuning on OR-related data. The results show that this kind of "layman" guidance is ineffective.
> > 2. How does majority voting (filtered null) work?
> >
> >    Thank you for pointing this out; we will clarify this in the paper. "Majority Voting (filtered null)" extracts the numerical objective value from all code outputs, discards failed ("null") runs, and selects the most frequent value among the valid results as the final answer.

---

> ### Author Response · Authors · 2025-11-23
>
> Q6. Did you try any out of distribution benchmarks?:
> > We focus exclusively on the specialized OR domain; therefore, the model’s performance in general-purpose scenarios may be limited, as its strengths are specifically tailored to tasks within area.
>
> Q7. How would expanding the dataset help to create credible baselines?：
> > The community currently lacks such a public dataset, so we pioneered the first one. Therefore, the core purpose of expanding it is to pave the way for future research and foster progress across the entire field.
>
> Q8. Appendix F:
> > 1.  Critic Prompt: seems only for the full solution
> >
> >     The prompt is unified by design; it can critique either a single step or the full solution, as it only evaluates the parts provided in the input.
> > 2. Can we actually call this a Process Reward Model, if it does not look at individual reasoning steps?
> >
> >    Yes, it is a Process Reward Model precisely because it does evaluate individual reasoning steps. It separately assesses the key logical stages of the solution, including the variables, the constraints, and the code, which is the definition of process level supervision.
> > 3. What is difference between the two prompts on page 24? Where are they applied?
> >
> >    The prompts separate the task into two stages: the first generates the mathematical model from the problem, and the second generates the executable code from that model.
>
> We hope these clarifications have thoroughly addressed all of your questions and concerns.

---

### Official Review · Reviewer_smtZ · 2025-11-01

**Soundness:** 3
**Presentation:** 3
**Contribution:** 3
**Rating:** 8
**Confidence:** 3

**Summary:**

This paper introduces the first Process Reward Model tailored for Operations Research (OR-PRM), aiming to explore the potential of Large Language Models (LLMs) in this complex reasoning domain. The authors surprisingly find that direct training of a PRM on existing mainstream OR datasets yields weak performance. Through systematic analysis, they identify the primary bottleneck as data quality, noting that over 30% of existing annotations are severely flawed. To address this, the authors collect all existing synthetic datasets and employ a carefully designed filtering pipeline to construct a high-quality seed dataset, which is then used for model training and evaluation. This work not only lays the foundation for applying LLMs in the OR field but also provides a crucial warning and proposed improvement strategy for the quality of current OR algorithm datasets.

**Strengths:**

1. This is the first application of the Process Reward Model paradigm to algorithmic problems in the field of OR. OR is a domain highly dependent on structured reasoning and complex algorithms, making the integration of LLMs highly valuable for research and application.

2. The authors conduct a systematic analysis that clearly identifies a severe annotation quality issue (over 30% defects) in mainstream OR datasets. This is a significant contribution and warning to the wider community, as identifying data bottlenecks is a crucial step for field advancement.

3. The authors do not avoid the data quality issue but instead proactively address it by collecting existing synthetic datasets and applying a "carefully designed filtering pipeline" to build a high-quality seed dataset. This strategy of solving the problem at the data source is commendable.

4. If OR-PRM proves effective, it will greatly simplify the modeling and solving process for OR problems, providing new avenues for automated algorithm discovery and solution.

**Weaknesses:**

No serious weakness.

**Questions:**

None

---

### Meta-Review · Area_Chair_usYh · 2026-01-07

**Summary:**

1) Reviewers raised concerns about novelty attribution and motivation, specifically whether the paper over-claims originality in identifying severe data quality issues in existing OR datasets, given prior related work.

2) There were concerns about dataset transparency, evaluation clarity, and potential circularity in the pipeline, including missing dataset statistics, unclear baseline definitions, and reliance on strong LLMs for both data construction and verification.

3) Reviewers also questioned the generality of the approach, noting limited out-of-distribution evaluation, incomplete comparisons to recent PRM variants, and insufficient factorization of the pipeline to isolate the contribution of the generative PRM component.

**Reviewer Concerns:**

1) This concern was largely addressed in the rebuttal. The authors clarified that while prior work established general awareness of data quality issues, their three-stage filtering pipeline is driven by an independent systematic audit finding that over 30% of samples in major OR datasets contain critical flaws. They committed to clearly separating prior observations from their own validation in the main text and to adding detailed audit statistics, which resolves the ambiguity pending revision.

2) These concerns were substantially addressed. The authors committed to adding comprehensive dataset statistics, clarified all evaluation protocols and baseline definitions, and provided manual audit results. While reliance on strong LLMs remains a limitation, the rebuttal clarifies how auditing, error checking, and model roles are handled in the pipeline.

3) These concerns remain only partially addressed. The authors acknowledge the lack of out-of-distribution evaluation, limited comparison to recent PRM variants, and incomplete factorized ablations, defer them to future work. While these are valid limitations, they are framed as scope constraints rather than fundamental flaws. In my opinion, they do not constitute an issue with regard to presenting the paper (as weak accept) at ICLR.

I would really encourage the authors to include all promised clarifications in the final paper. Also those made during the rebuttal phase.

**Reviewer Scores:**

Hard to tell, as there was no interaction at all. But I can see the following potential changes:

Reviewer gX4v: 2 -> 4

Reviewer xTZk: 4 -> 6

---

### Decision · Program_Chairs · 2026-01-26

Accept (Poster)